health and disease and epidemiology/ biomathematics/mathematical modelling

mathematical model, COVID-19, SARS-CoV-2, vaccines, non-pharmaceutical and pharmaceutical interventions, face masks and respirators

**Author for correspondence:**
Calistus N. Ngonghala
e-mail: calistusnn@ufl.edu

# Assessing the impact of widespread respirator use in curtailing COVID-19 transmission in the USA

Calistus N. Ngonghala[1,2], James R. Knitter[3], Lucas Marinacci[4], Matthew H. Bonds[5] and Abba B. Gumel[6,7]

[1]Department of Mathematics, and [2]Emerging Pathogens Institute, University of Florida, Gainesville, FL, USA
[3]Division of Pulmonary, Allergy, Critical Care, and Sleep Medicine, College of Medicine, University of Arizona, Tucson, AZ, USA
[4]Division of Cardiovascular Medicine, Beth Israel Deaconess Medical Center and Harvard Medical School, Boston, Massachusetts, USA
[5]Department of Global Health and Social Medicine, Harvard Medical School, Boston, MA, USA
[6]School of Mathematical and Statistical Sciences, Arizona State University, Tempe, AZ, USA
[7]Department of Mathematics and Applied Mathematics, University of Pretoria, Pretoria, South Africa

CNN, 0000-0002-8441-9495; ABG, 0000-0002-8413-1248

Dynamic models are used to assess the impact of three types of face masks (cloth masks, surgical/procedure masks and respirators) in controlling the COVID-19 pandemic in the USA. We showed that the pandemic would have failed to establish in the USA if a nationwide mask mandate, based on using respirators with moderately high compliance, had been implemented during the first two months of the pandemic. The other mask types would fail to prevent the pandemic from becoming established. When mask usage compliance is low to moderate, respirators are far more effective in reducing disease burden. Using data from the third wave, we showed that the epidemic could be eliminated in the USA if at least 40% of the population consistently wore respirators in public. Surgical masks can also lead to elimination, but requires compliance of at least 55%. Daily COVID-19 mortality could be eliminated in the USA by June or July 2021 if 95% of the population opted for either respirators or surgical masks from the beginning of the third wave. We showed that the prospect of effective control or elimination of the pandemic using mask-based strategy is greatly enhanced if combined with other non-pharmaceutical interventions (NPIs) that significantly reduce the baseline community transmission. By slightly modifying the model to include the effect of a vaccine against

COVID-19 and waning vaccine-derived and natural immunity, this study shows that the waning of such immunity could trigger multiple new waves of the pandemic in the USA. The number, severity and duration of the projected waves depend on the quality of mask type used and the level of increase in the baseline levels of other NPIs used in the community during the onset of the third wave of the pandemic in the USA. Specifically, no severe fourth or subsequent wave of the pandemic will be recorded in the USA if surgical masks or respirators are used, particularly if the mask use strategy is combined with an increase in the baseline levels of other NPIs. This study further emphasizes the role of human behaviour towards masking on COVID-19 burden, and highlights the urgent need to maintain a healthy stockpile of highly effective respiratory protection, particularly respirators, to be made available to the general public in times of future outbreaks or pandemics of respiratory diseases that inflict severe public health and socio-economic burden on the population.

## 1. Introduction

Since its emergence at the end of December 2019, the novel coronavirus pandemic (COVID-19) has been causing devastating public health and socio-economic burden around the world. The COVID-19 pandemic has, as of 27 March 2021, accounted for over 126.5 million confirmed cases and over 2.77 million reported deaths globally [1,2]. The USA has disproportionately contributed about 30 million reported cases, accounting for more than 25% of the total global burden, and over 548 000 reported COVID-19 deaths [2,3]. In fact, the US Centers for Disease Control and Prevention (CDC) estimated that the true number of COVID-19 cases in the USA by the end of 2020 exceeds 80 million. This figure account for the fact that no wide-scale, random testing has been implemented to detect asymptomatic and pre-symptomatic cases, of which 4.1 million were hospitalized [4]. The estimated economic impact of COVID-19 in the USA (measured in terms of aggregated mortality, morbidity, mental health conditions and direct economic losses) is as high as 16 trillion USD [5–9]. Due to this enormous public health and socio-economic burden, the use of all possible control and mitigation strategies is necessary to prevent further illness and death.

In response, both pharmaceutical interventions, including vaccines, monoclonal antibodies and small-molecule drugs, as well as non-pharmaceutical interventions (NPIs), such as social distancing, hand washing and mask wearing, have been used to slow the spread and limit the impact of the pandemic. COVID-19 vaccines are now being deployed throughout the world, most widely in high-income countries [10]. However, the roll-out of these vaccines is being hampered by severe supply chain constraints even among high-income countries, leaving many individuals susceptible to infection and associated morbidity and mortality. This situation is especially pronounced in low- and middle-income countries (LMICs) that face real difficulties in competing for and acquiring scarce vaccine supplies. This disparity is projected to leave up to one quarter of the world's population without COVID-19 vaccines until at least 2022 [11]. Even among those who ultimately have access, an estimated 32% of the global population is hesitant or unwilling to receive a COVID-19 vaccine, posing a significant challenge to the goal of achieving widespread vaccine-induced immunity against the pandemic [12]. Additionally, COVID-19 variants of concern are continuing to emerge that reduce the effectiveness of existing vaccines [13], which could drive the need for continuous development of reformulated vaccines. As such, while vaccines are a powerful tool, they represent only one tool in the armamentarium of all available control measures against the pandemic.

Of the non-pharmaceutical interventions, masking has been among the most contentious in specific countries, including the USA, due in part to politicization and conflicting early recommendations. Nevertheless, masking remains a crucial intervention, even in communities or populations with widespread vaccine deployment. Especially for those who are hesitant or unwilling to be vaccinated but who are willing to wear a mask, continued masking helps prevent disease transmission in otherwise susceptible individuals. Masking also presents a partial solution to variants of concern. Whereas variants have potential to evade vaccines, masks present a broad-spectrum physical barrier that is relatively insensitive to specific variants. With regard to variants of concern that demonstrate increased transmissibility, some public health experts have suggested that the use of higher-efficacy respiratory protection may be necessary to maintain similar levels of control compared with wild-type SARS-CoV-2.

As masking remains a vital strategy in the control of the COVID-19 pandemic, the efficacy of various mask options available to the general public should be considered in order to optimize protection against disease transmission. Broadly, several categories of respiratory protection exist. These include cloth masks, surgical or procedure masks, and respirators, which are defined as tight-fitting respiratory protective devices that meet or exceed the National Institute for Occupational Safety and Health (NIOSH) N95 standard [14–16]. Cloth masks and surgical or procedure masks are considered to be loose fitting face coverings and are designed to be used as source control. Their primary purpose is preventing the spread of disease from infected wearers to those around them, and not the protection of the wearer from others. Due to the unregulated and highly heterogeneous nature of cloth mask design and construction, efficacy estimates vary widely, ranging from 0 to 50% [16,17]. Like cloth masks, surgical masks and procedure masks are also designed to be loose fitting and act as source control but are made from materials that have a theoretical filtration efficiency of up to 85% [18]. Due to their lack of a tight seal, the real-world filtration efficiency of the surgical or procedure masks can be much lower [16]. In contrast to cloth and surgical masks, most respirators are designed to fit tightly to the face and represent true respiratory protection, protecting the wearer from respiratory hazards around them. Examples of NIOSH-certified respirators include N95, N99, N100, R95, P95, P99 and P100 particulate filtering face-piece respirators [14,15]. Estimates for the efficiency of filtering face-piece respirators are nearly 100% for charged biological particles such as respiratory aerosols [15,16,18]. In all cases, poor fit can impair efficiency, leading to decreased real-world fitted filtration efficiency ratings that are lower than theoretical maximums as determined solely by testing of the filtration material [19]. This effect is especially important when considering the differences between loose-fitting (cloth and surgical mask) and tight-fitting (respirator) devices. Loose-fitting devices are not designed to form a seal and thus enable significant filter bypass. This results in very low real-world fit factors in the low single-digits [19]. In comparison, respirators are able to form a tight seal and, when properly fitted, are able to achieve fit factors at and above 100, which represents a 100-fold reduction in the concentration of particles inside the device compared with particles outside of the device [20]. Due to the highly protective nature of properly fitting respirators, healthcare workers have used these devices for protection from SARS-CoV-2 throughout the pandemic.

Since the onset of the pandemic, the prevailing public health recommendations with regard to masking have been that cloth and surgical masks should be used by members of the general public partly due to being widely available. Recently, however, calls have been made to the CDC by public health and respiratory protection experts to change their public health guidance and to begin recommending the use of higher-efficacy respiratory protection, such as respirators, for individuals outside of healthcare roles [21]. These individuals could include those at high risk of acquiring SARS-CoV-2 infection or from populations that have suffered disproportionately, including essential workers such as those at grocery stores and meat-packing plants; people of specific races and ethnic background, including Black, Indigenous, Latino and other people of colour who have been heavily impacted by the pandemic; older individuals; and those with underlying medical conditions placing them at increased risk of severe illness and death, including cancer, chronic kidney disease, chronic obstructive pulmonary disease (COPD), Down syndrome, heart disease, transplant recipients, obesity, pregnancy, sickle cell disease, smoking and diabetes [22]. The rationale for doing so stems from the theoretical improved protection that respirators could provide to those at high risk. As vaccination for many of these individuals is still months away even in high-income countries, and up to a year or more away for those in LMICs, the recommendation to use improved respiratory protection could play a role in lowering COVID-19 disease transmission and associated morbidity and mortality. As current variants of concern circulate and new concerning variants are likely to arise in the future, maximizing the available non-pharmaceutical interventions could also serve in a vaccine-sparing and therapeutic-sparing capacity by further limiting the ability of variants to emerge due to selective pressures against specific therapeutic monoclonal antibodies and vaccines.

The impact of the use of respiratory protection with efficacy far exceeding that of cloth and surgical masks for members of the general public is currently not known. Previous modelling studies have focused on the role of lower-efficacy face coverings, such as cloth and surgical masks, when adopted by a large majority of the population [17,23–25]. What is not clear is the potential role that highly efficacious respiratory protection could serve outside of healthcare and what levels of use in the population may be necessary to project significant changes compared with strategies employing lower-efficacy masks. As populations use a combination of varying categories of face coverings, it is

also not known how combinations of different types of respiratory protection could impact disease transmission. Using mechanistic mathematical modelling, we studied the role of high-efficacy respiratory protection when compared with lower-efficacy solutions commonly suggested through public health messaging. This study clarifies the role that high-efficacy respiratory protection, such as respirators, could serve in decreasing COVID-19 disease transmission and speeding an end to the pandemic, as well as highlighting the implications of potential new strategies to fight future epidemics and pandemics.

# 2. The basic model

## 2.1. Formulation of basic model

We start with formulating a basic mathematical model for the transmission dynamics of COVID-19 in a population where a certain proportion habitually wear face masks (cloth, surgical or respirators) in public. The basic model considers the total population at time $t$, denoted by $N(t)$, as a single group that is subdivided into epidemiological compartments of individuals who are susceptible ($S(t)$), exposed/latent ($E(t)$), pre-symptomatically infectious ($P(t)$), symptomatically infectious (i.e. individuals who survived the incubation period and exhibit moderate to severe clinical symptoms of the disease, $I(t)$), asymptomatically infectious ($A(t)$), hospitalized ($H(t)$) and recovered $R(t)$, so that $N(t) = S(t) + E(t) + P(t) + I(t) + A(t) + H(t) + R(t)$. 'Exposed' individuals are those who are already infected with the disease but are not yet infectious (i.e. they have latent infection). While pre-symptomatically infectious individuals are those who transmit the disease after the latent period, but before the end of the incubation period, asymptomatically infectious individuals are those who survived the incubation period but display only mild or no clinical symptoms of the disease.

The basic model is given by the following deterministic system of nonlinear differential equations for the rate of change of each of the epidemiological compartments (where a dot represents differentiation with respect to time):

$$\left.\begin{aligned}
\dot{S} &= -(1 - c_m \varepsilon_m)\beta\left(\frac{\eta_P P + \eta_I I + \eta_A A + \eta_H H}{N}\right)S, \\
\dot{E} &= (1 - c_m \varepsilon_m)\beta\left(\frac{\eta_P P + \eta_I I + \eta_A A + \eta_H H}{N}\right)S - \sigma_E E, \\
\dot{P} &= \sigma_E E - \sigma_P P, \\
\dot{I} &= r\sigma_P P - (\phi_I + \gamma_I + \delta_I)I, \\
\dot{A} &= (1 - r)\sigma_P P - \gamma_A A, \\
\dot{H} &= \phi_I I - (\gamma_H + \delta_H)H, \\
\dot{R} &= \gamma_I I + \gamma_A A + \gamma_H H.
\end{aligned}\right\}
\tag{2.1}$$

In the basic model (2.1), $\beta$ is the overall community transmission rate, $0 < c_m < 1$ is the proportion of individuals who wear faces masks in public, $0 < \varepsilon_m < 1$ is the protective efficacy of face masks to prevent the transmission of infection to a susceptible individual and $\eta_j$ (with $j \in \{P, I, A, H\}$) represents the modification parameter for the infectiousness of the pre-symptomatic ($\eta_P$), symptomatic ($\eta_I$), asymptomatic ($\eta_A$) and hospitalized ($\eta_H$) infectious individuals. The parameter $\sigma_E$ is the progression rate of exposed (i.e. newly infected individuals but not yet infectious) individuals to the pre-symptomatic class; $\sigma_P$ is the rate at which pre-symptomatic individuals progress to either the symptomatic class or the asymptomatic class at the end of the pre-symptomatic period (i.e. after surviving the incubation period), with $0 < r < 1$ as the proportion of the pre-symptomatic individuals who show clinical symptoms of the disease at the end of the pre-symptomatic period (the remaining proportion, $1 - r$, do not show clinical symptoms at the end of the pre-symptomatic period, and are moved to the asymptomatically infectious class). Symptomatic infectious individuals are hospitalized at a rate $\phi_I$. Individuals in the symptomatic, asymptomatic and hospitalized infectious classes recover at a rate $\gamma_I$, $\gamma_A$ and $\gamma_H$, respectively. Symptomatic and hospitalized individuals suffer disease-induced mortality at a rate $\delta_I$ and $\delta_H$, respectively. The basic model forms the foundation for our estimation of unknown parameters, particularly the overall community transmission rate ($\beta$), the mortality rates ($\delta_I$ and $\delta_H$) and the recovery rates ($\gamma_I$, $\gamma_A$ and $\gamma_H$). The state

variables and parameters of the basic model (2.1) are described in table S1 of the electronic supplementary material.

In the formulation of the basic model (2.1), it is assumed that the total population is well-mixed (i.e. every member of the community is equally likely to mix with, and acquire infection from, every other member of the community) and the average waiting time in each epidemiological compartment is exponentially distributed [26]. It is also assumed that hospitalized individuals can transmit infection (at the rate $\eta_H\beta$; this assumption can be relaxed, if relevant, by setting $\eta_H = 0$). Recovery from COVID-19 is assumed to confer permanent immunity against future SARS-CoV-2 infections [23–25,27,28]. Although a number of COVID-19 variants have been identified in Brazil, South Africa, the UK and the USA [29–32], they are not included in this model due to insufficient data on how they might interact with the predominant strain. We define a parameter, $0 \leq c_r < 1$, is the proportion of individuals in the community who observe other additional forms of non-pharmaceutical interventions for reducing community transmission, such as through social distancing. The control reproduction number of the basic model (2.1), denoted by $\mathbb{R}_c$, is given by [33,34] $\mathbb{R}_c = \mathbb{R}_P + \mathbb{R}_I + \mathbb{R}_A + \mathbb{R}_H$, where, $\mathbb{R}_P = (1 - \varepsilon_m c_m)\eta_P\beta/\sigma_P$, $\mathbb{R}_I = (1 - \varepsilon_m c_m)\eta_I\beta r/(\phi_I + \gamma_I + \delta_I)$, $\mathbb{R}_A = (1 - \varepsilon_m c_m)\eta_A\beta(1 - r)/\gamma_A$ and $\mathbb{R}_H = (1 - \varepsilon_m c_m)\eta_H\beta r\phi_I/(\phi_I + \gamma_I + \delta_I)(\gamma_H + \delta_H)$ represent, respectively, the constituent control reproduction numbers associated with disease transmission by the pre-symptomatic infectious ($\mathbb{R}_P$), symptomatically infectious ($\mathbb{R}_I$), asymptomatically infectious ($\mathbb{R}_A$) and hospitalized individuals ($\mathbb{R}_H$) [23–25]. The control reproduction number ($\mathbb{R}_c$) represents the average number of new COVID-19 cases generated by a typical infected individual introduced into a population where a certain proportion ($0 < c_m < 1$) habitually wears face masks in public. The implication of this threshold quantity is that an outbreak will occur in the community if the value of $\mathbb{R}_c$ exceeds unity, and no significant outbreak occurs otherwise [24,25].

## 2.2. Calibration and parameter estimation of basic model

The basic model (2.1) contains 16 parameters, the values of 10 of which ($c_m$, $\varepsilon_m$, $\eta_P$, $\eta_I$, $\eta_A$, $\eta_H$, $\sigma_E$, $\sigma_P$, $r$ and $\phi_I$) are known from the literature, while the values of the remaining six ($\beta$, $\gamma_I$, $\gamma_A$, $\gamma_H$, $\delta_I$ and $\delta_H$) are unknown. Among the known parameters that are shown to be critical to the disease dynamics (as shown in §2.3) are the compliance in mask usage ($c_m$) and the protective efficacy of masks ($\varepsilon_m$). While the value of the mask compliance parameter is adapted from [24] and the prevailing compliance level in the USA at the time of this study, the value of the mask efficacy parameter is estimated from data provided in laboratory and empirical evaluations of various forms of masks and respirators [18,35,36]. Furthermore, the contour plot depicted in figure 3 captures the robust and full-scale impact (and sensitivity) of these two parameters on the disease dynamics (as measured by the control reproduction number of the model). In other words, the contour plot allows us to evaluate the sensitivity of these parameters (for all possible values in their respective parameter space) on the reproduction number of the model.

The basic model (2.1) will be fitted with real data to obtain realistic estimates of the unknown parameters of the model. Specifically, we use the cumulative COVID-19 mortality data for the USA (since this is more reliable than case data [24]) for three different periods (22 January to 30 June 2020, 1 July to 11 October 2020 and 12 October 2020 to 11 March 2021), corresponding to the three observed COVID-19 pandemics waves in the USA, to fit the single group model (2.1). In order to overcome the issue of avoidable errors introduced by fitting cumulative data to deterministic models [37], we also use new daily mortality data for the same period to check the quality of our fit. Based on the fitting, we obtain estimates for the community transmission rate ($\beta$), the mortality rates for symptomatic infectious and hospitalized individuals ($\delta_I$ and $\delta_H$, respectively), and the recovery rates of symptomatic infectious, asymptomatic infectious and hospitalized individuals ($\gamma_I$, $\gamma_A$ and $\gamma_H$, respectively) for each of the three pandemic waves. The fitting is achieved by using a simple nonlinear regression process, in which the sum of the squared differences between the observed cumulative mortality data for the USA and the cumulative mortality from the model output (i.e. $\int_0^T (\delta_I I + \delta_H H)\,dt$, where $T$ is the time duration for the simulations) is minimized and a bootstrapping approach is used to compute the 95% credible intervals for the estimated parameters [23,38,39].

Time-series plots illustrating the fit, and how well the output of the model matches the daily data for the same time period, are depicted in figure 1a,b, respectively. The estimated values of the parameters obtained from fitting the basic model are tabulated in table 2.

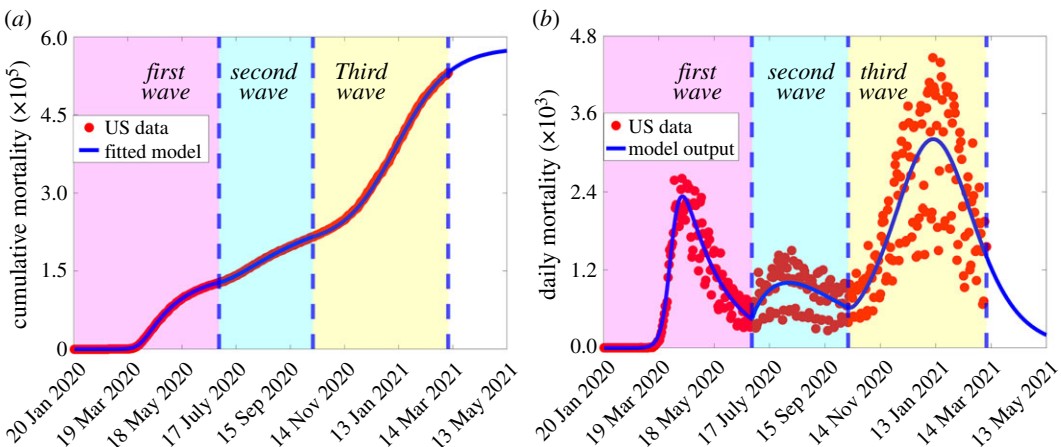

**Figure 1.** (*a*) Time-series plot of the observed/reported cumulative mortality data for the USA (red dots) and predicted cumulative mortality for the USA generated from the basic model (2.1). (*b*) Simulations of the basic model (2.1) using the fixed parameter values in table 1 and the estimated parameter values in table 2, illustrating the three COVID-19 waves observed in the USA during the periods from 20 January to 30 June 2020 (first pandemic wave), 1 July to 11 October 2020 (second pandemic wave), and 12 October 2020 to 11 March 2021 (third pandemic wave). Dashed vertical blue lines demarcate the pandemic waves.

**Table 1.** Fixed baseline parameter values of the basic model (2.1). Although the baseline value of the mask compliance parameter ($c_m$) is set at 0.50, we use different baseline values for the first and second pandemic waves. In particular, we set the baseline values for the first and second waves to $c_m \approx 0.20$ and $c_m \approx 0.35$, respectively, which correspond to the approximate value of the mask compliance during the first and second waves of the COVID-19 pandemic in the USA.

| parameter | value | source |
|---|---|---|
| $\eta_A$ | 1.00 | adapted from [40] |
| $\eta_P$ | 0.75 | adapted from [40] |
| $\eta_I$ | 0.50 | adapted from [40] |
| $\eta_H$ | 0.15 | adapted from [40] |
| $\varepsilon_m$ | 0.50 | estimated from [18,35,36] |
| $\varsigma_m$ | 0.50 | adapted from [24] |
| $r$ | 0.20 | [41,42] |
| $\sigma_E$ | 0.40 | [43,44] |
| $\sigma_P$ | 0.40 | [43–45] |
| $\Phi_I$ | 0.17 | [46] |

## 2.3. Simulation results from the basic model

The basic model (2.1) is now simulated to assess the population-level impact of face masks usage in the community. For simulation purposes, we consider four categories of face masks, namely cloth mask (with estimated efficacy 30%), improved cloth masks or poorly fitted surgical mask (with estimated efficacy of 50%), properly fitted surgical mask (with estimated efficacy of 70% [16,18,35]), and properly fitted respirators, with estimated efficacy of 95%. The impact of combining the face masks use strategy with an additional adoption of NPIs that target reduction in community transmission of COVID-19 (such as social distancing, self-isolation, avoiding travels and/or public gatherings, avoiding indoor dining in restaurants, etc.), as measured by the parameter $c_r$, will also be assessed. The simulations will be carried out using the fixed baseline parameter values presented in table 1 and the estimated baseline parameter values as presented in table 2. It is worth mentioning that, for these baseline values of the parameters of the basic model (2.1), the values of the associated control reproduction number $\mathbb{R}_c$) corresponding to the first, second and third waves of the COVID-19 pandemic in the USA are $\mathbb{R}_c = 3.1$, $\mathbb{R}_c = 1.08$ and $\mathbb{R}_c = 1.2$, respectively (confirming the devastating nature of the first wave, in relation to the other two waves recorded so far).

**Table 2.** Estimated (fitted) baseline parameter values and confidence intervals (CIs) for the basic model (2.1) using COVID-19 mortality data for the USA for the period from (a) 20 January to 30 June 2020 (first pandemic wave), (b) 1 July to 11 October 2020 (second pandemic wave) and (c) 12 October 2020 to 11 March 2021 (third pandemic wave). The unit of each of the parameters is day$^{-1}$.

| parameter | value | CI |
|---|---|---|
| **(a) first wave** | | |
| $\beta$ | 0.723402 | [0.711360, 0.739537] |
| $\gamma_I$ | 0.191951 | [0.176000, 0.289251] |
| $\gamma_A$ | 0.382469 | [0.346000, 0.498512] |
| $\gamma_H$ | 0.024724 | [0.014247, 0.035168] |
| $\delta_I$ | 0.000065 | [0.000024, 0.000089] |
| $\delta_H$ | 0.000113 | [0.000106, 0.000118] |
| **(b) second wave** | | |
| | 0.3045 | [0.2547, 0.3524] |
| | 0.1986 | [0.1860, 0.2898] |
| | 0.4859 | [0.3760, 0.6864] |
| | 0.0271 | [0.0221, 0.0331] |
| | 0.0001 | [0.0000, 0.0002] |
| | 0.0004 | [0.0003, 0.0005] |
| **(c) third wave** | | |
| | 0.4332 | [0.4152, 0.4570] |
| | 0.2989 | [0.2043, 0.3757] |
| | 0.5091 | [0.4629, 0.5798] |
| | 0.1698 | [0.0395, 0.2778] |
| | 0.0037 | [0.0025, 0.0069] |
| | 0.0066 | [0.0000, 0.0094] |

### 2.3.1. Assessing the impact of early implementation of mask use strategy

The first set of simulations we carried out are for assessing the impact of early implementation of mask use strategy in the USA. Specifically, we simulated the scenario where only one of the aforementioned four categories of face masks is adopted in the community. For these simulations, we used the baseline values of the fixed and estimated parameters corresponding to the first wave of the pandemic (tabulated in tables 1 and 2). The results obtained, depicted in figure 2, show that the early implementation of universal face mask use strategy, with compliance of 80%, resulted in a dramatic reduction in the burden of the pandemic, particularly if high-quality masks, notably surgical masks and respirators, are used. In particular, if 80% of Americans were wearing respirators starting from 15 March 2020 (which was almost two months after the index case on 20 January 2020, or 4 days after the WHO declared COVID-19 to be a global pandemic), up to 84% of the nearly 128 000 baseline cumulative deaths recorded by 30 June 2020 would have been prevented (figure 2a, green curve). On the other hand, if surgical masks were chosen instead, the reduction in the baseline cumulative mortality by 30 June 2020 would have been 43% (figure 2a, cyan curve). The reductions corresponding to the use of improved cloth mask or basic cloth masks only are 18% (figure 2a, magenta curve), or 5% (figure 2a, gold curve), respectively.

Much higher reduction in the baseline cumulative mortality will be recorded if the universal masking strategy was implemented earlier than 15 March 2020. For example, if the universal masking strategy was started on 18 February 2020 (i.e. a month after the first reported case in the USA), the pandemic would have hardly taken off if the (80%) masks-wearing proportion chose to wear respirators only (figure 2b, green curve). On the other hand, choosing to wear cloth mask only or improved cloth mask

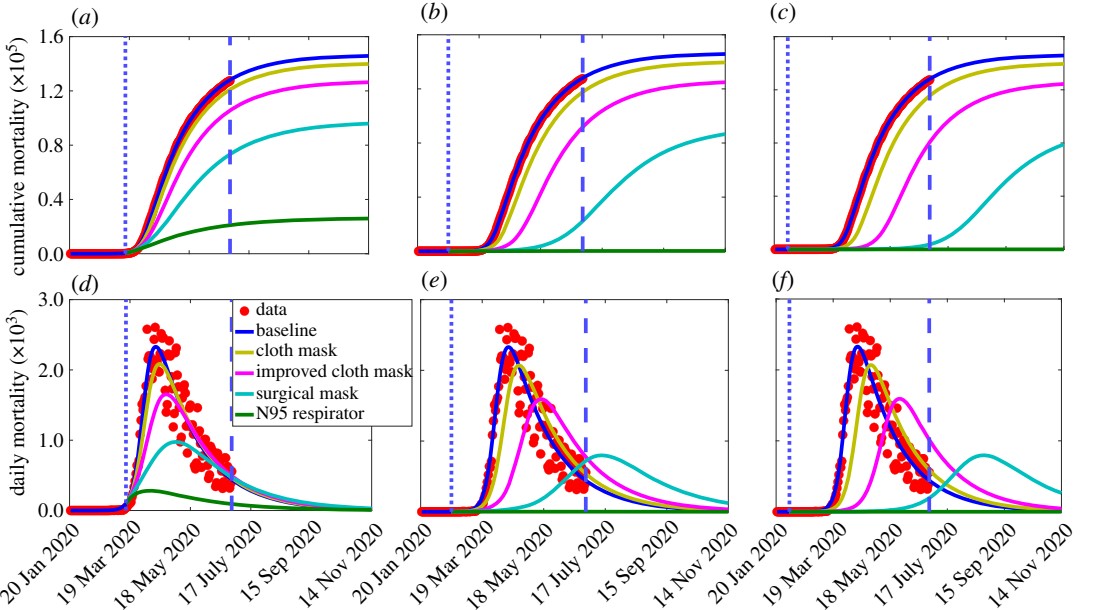

**Figure 2.** Simulations of the basic model (2.1) to assess the impact of wide adoption of face mask wearing policy (assuming 80% compliance) on the cumulative COVID-19 mortality (*a–c*) and daily COVID-19 mortality (*e–f*) in the USA, starting from the onset of the first pandemic wave (20 January 2020). Scenarios include cloth masks only (with estimated efficacy of 30%), improved cloth masks only (with estimated efficacy of 50%), surgical masks only (with estimated efficacy of 70%), or respirators only (with efficacy of 95%), The other parameter values used in the simulations are given in table 1 and the first wave parameter values in table 2(*a*). The dotted vertical blue line depicts the onset of the face mask use strategy, while the dashed vertical blue line depicts the end of the first pandemic wave: (*a,d*) 15 March 2020 (i.e. 4 days after the World Health Organization declared COVID-19 to be a global pandemic), (*b,e*) 18 February 2020 (i.e. one month after the US index case), and (*c,f*) 2 February 2020 (i.e. two weeks after the US index case).

only or (even) surgical masks only would have resulted in a sizable epidemic (figure 2*b*, gold, magenta and cyan curves). Similar results are obtained if the mask use strategy was implemented two weeks after the index case in the USA (figure 2*c*). Plots for daily mortality are depicted in figures 2*d–f*, showing similar trends as the plots for the cumulative mortality (figure 2*a–c*). In all of these simulations, the COVID-19 pandemic will not have taken off in the USA if the assumed 80% of the mask-wearing population chose to wear respirators (or equivalent) only. Furthermore, these simulations show that choosing properly fitted surgical masks would also have made significant impact in reducing COVID-19 mortality, but certainly nowhere as effective as the case where respirators were preferred. The respirators and surgical masks are, of course, far more effective than using the improved or regular cloth mask.

### 2.3.2. Assessing the combined impact of face masks usage and other non-pharmaceutical interventions

Additional simulations of the basic model (2.1) are carried out to assess the combined impact of face masks usage and other NPIs. It should also be emphasized that, since the model (2.1) was parametrized using COVID-19 data from the onset of the outbreak in the USA (i.e. using data from 20 January 2020 to mid-March 2021), the combined effects of other NPIs that resulted in a reduction in community transmission of COVID-19 (such as social distancing, travel restrictions, avoiding large gatherings and indoor dining in restaurants) are already embedded into the results/data. That is, our model parameter calibration includes some baseline level of the other NPIs implemented in the community. To assess the impact of the additional levels of these other NPIs on the burden of the pandemic (from their baseline levels), we multiply the effective community transmission rate ($\beta$) by the fraction $1 - c_r$, where $0 < c_r \leq 1$ is a measure of the additional reduction in the community transmission rate. Since our predictions are based on the estimated parameters for the third pandemic wave (i.e. the parameter values estimated using the data from 12 October 2020 to mid-March 2021), the additional reduction in community transmission due to the implementation of other NPIs (modelled by the parameter $c_r$) starts on 12 October 2020 and is maintained through the rest of the simulation (or prediction) period (for these, and future simulations in this paper, we used the fixed and estimated values of the parameters corresponding to the third wave of the pandemic).

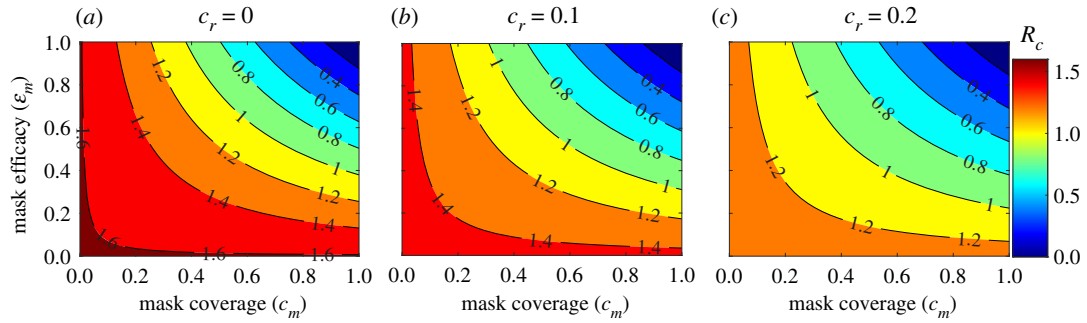

**Figure 3.** Contour plot of the control reproduction number ($\mathbb{R}_c$) of the basic model (2.1), as a function of the face masks compliance ($c_m$) and efficacy ($\varepsilon_m$). (*a*) No reduction in community transmission from baseline due to the implementation of other NPIs (i.e. $c_r = 0.0$). (*b*) 10% reduction in community transmission due to implementation of other NPIs (i.e. $c_r = 0.1$). (*c*) 20% reduction in community transmission due to the implementation of other NPIs (i.e. $c_r = 0.2$). The other parameter values used in the simulations are as given in tables 1 and 2(*c*).

Figure 3 depicts a contour plot of the control reproduction number of the model ($\mathbb{R}_c$) as a function of face masks compliance ($c_m$) and efficacy ($\varepsilon_m$) for various levels of increases in baseline values of reduction in community transmission due to other NPIs ($c_r$). The figure indicates a maximum control reproduction number during the simulation period (12 October 2020 to 11 March 2021) of $\mathbb{R}_c = 1.6$ (with a baseline value of $\mathbb{R}_c = 1.2$; signifying a continuing trajectory of disease spread in the USA, since $\mathbb{R}_c > 1$ during the third wave. Furthermore, this figure shows that $\mathbb{R}_c$ decreases with increasing values of face masks compliance and efficacy. In the absence of additional reduction in community transmission due to the implementation of other NPIs (i.e. $c_r = 0$), our simulations (figure 3*a*) show that if individuals in the community choose to use cloth masks only (with estimated efficacy of 30%), the control reproduction number can never be brought to a value less than unity (hence, the disease will continue to spread in the population).

On the other hand, if the American populace choose to use improved cloth masks (with estimated efficacy of 50%) only, a compliance of at least 75% will be needed to bring the control reproduction number to a value less than one. The compliance level needed reduces to 55% if Americans choose, instead, to wear well-fitted surgical masks only. Finally, if respirators were adopted right from the beginning of the pandemic, this figure shows that only 40% compliance will be required to reduce the control reproduction number below unity. In other words, the pandemic can be effectively controlled (eliminated) if 40% of the American populace were wearing face masks of respirators quality right from the beginning of the pandemic outbreak. Furthermore, if the face masks use strategy is combined with other NPIs (i.e. if $c_r \neq 0$), our simulations show that the face masks compliance level needed to bring the control reproduction number to a value less than unity decreases significantly. For instance, if a 10% increase in reduction in community transmission due to other NPIs (from the baseline) can be achieved (i.e. if $c_r = 0.1$), our simulations show that the reproduction number can be brought to a value less than one using the low-quality cloth masks if compliance is near 100% (figure 3*b*). For this scenario, the use of improved cloth masks only would require compliance of 60%, while the use of surgical masks and respirators (or equivalent) only would require compliance of only 45% and 32%, respectively, to bring the control reproduction number below one. For the case where the implementation of other NPIs can lead to a decrease in community transmission by 20%, our simulations show that the use of low-quality cloth masks can bring the reproduction number to a value less than one if at least 75% of Americans wear them (figure 3*c*). The required compliance reduces to 45% if improved cloth masks are chosen instead. The compliance further reduces to 32% if surgical masks are preferred and to 23% if only respirators are used.

Additional simulations of the basic model are carried out to assess the impact of the face masks strategy on cumulative mortality for various levels of reduction in community transmission due to the implementation of other NPIs. Specifically, we simulated the scenario for COVID-19 dynamics in the USA, starting from 12 October 2020, where only cloth masks, improved cloth masks, surgical masks, or respirators are used. We consider 30 April 2021 (the end of the first 100 days of the new administration in the USA) as a reference point for projections. Figure 4 shows that, based on the baseline values of the fitted and fixed parameters in tables 1 and 2, about 570 000 cumulative deaths would be occur in the USA by 30 April 2021 if only improved cloth masks are used (figure 4*a–d*, blue

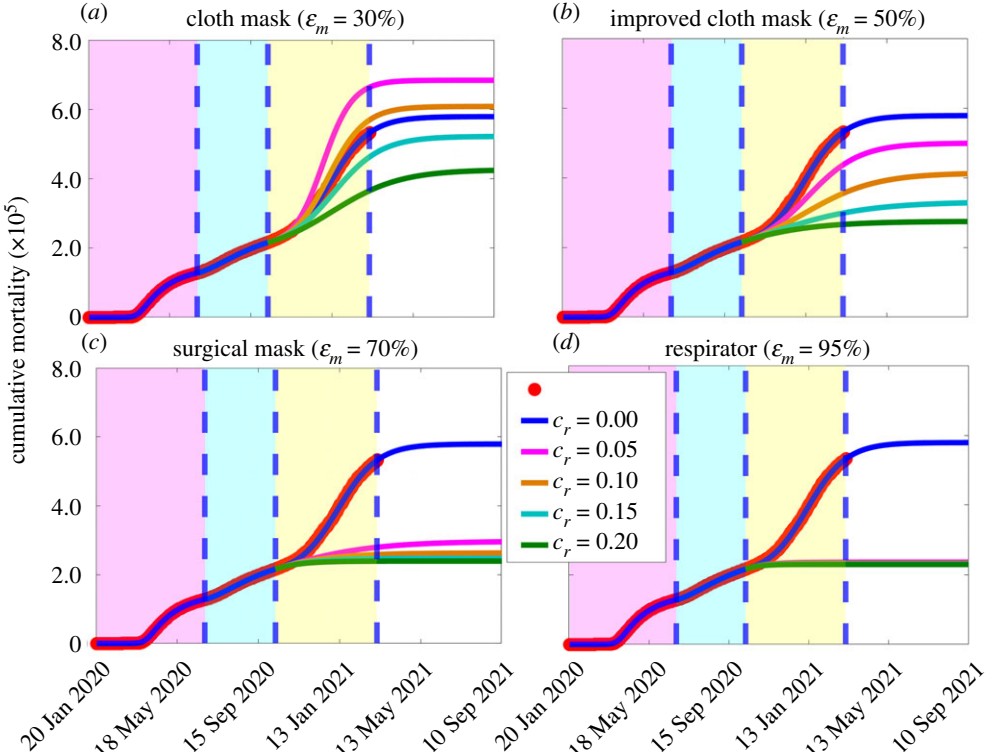

**Figure 4.** Simulations of the basic model (2.1) to assess the combined impact of face mask use strategy and reduction in community transmission due to other NPIs (implemented at the beginning of the third wave) on cumulative COVID-19 mortality in the USA (a) Only cloth masks are used. (b) Only improved cloth masks are used. (c) Only surgical masks are used. (d) Only respirator (or equivalent) are used. For the blue curves (baseline scenario) in figures (a–d), the mask efficacy is maintained at 50% (i.e. $\varepsilon_m = 0.5$). The other parameter values used in the simulations are given in tables 1 and 2. The shaded magenta and green regions represent the beginning and end of the first and second pandemic waves in the USA, respectively. The shaded yellow region represents the beginning of the third pandemic wave until 11 March 2021. Dashed vertical blue lines demarcate the three waves.

**Table 3.** Percentage increase (denoted by a '+' sign) or reduction (denoted by a '−' sign) in cumulative mortality by 30 April 2021 (in comparison with the 570 000 projected baseline value), if only cloth, improved cloth, surgical or respirators (or equivalent) mask is used in the USA, for various levels of reduction in community transmission due to the implementation of other NPIs implemented at the beginning of the third wave (as depicted in figure 4).

| mask type | $c_r = 0.05$ | $c_r = 0.10$ | $c_r = 0.15$ | $c_r = 0.20$ |
|---|---|---|---|---|
| cloth mask ($\varepsilon_m = 30\%$) | +20% | +5% | −11% | −30% |
| improved cloth mask ($\varepsilon_m = 50\%$) | −16% | −32% | −45% | −52% |
| surgical mask ($\varepsilon_m = 70\%$) | −50% | −54% | −57% | −58% |
| respirator or equivalent ($\varepsilon_m = 95\%$) | −58% | −59% | −59% | −60% |

curve). If implementing only a cloth mask strategy along with other NPIs that can reduce community transmission during the beginning of the third wave (12 October 2020) by only 5% (respectively, 10%) (i.e. $c_r = 0.05$ (respectively, $c_r = 0.10$)), about 112 000 (respectively, 32 000) deaths above the projected 570 000 baseline cumulative deaths by 30 April 2021 could be recorded (figure 4a, magenta curve) (respectively, figure 4a, gold curve). This represents approximately a 20% (respectively, 6%) increase in the projected baseline cumulative mortality by 30 April 2021. If the reduction in community transmission is increased to 15% (i.e. $c_r = 0.15$), then about 64 000 (11%) of the projected 570 000 cumulative mortality by 30 April 2021 would have been averted (figure 4a, cyan curve). About 30% (i.e. 170 000) of the projected mortality by 30 April 2021 would be prevented if NPIs could reduce community transmission by 20% during the beginning of the third wave (figure 4a, green curve). On

the other hand, if a combination of respirators (or equivalent) are used, the projected cumulative mortality by 30 April 2021 reduces dramatically (figure 4d). Here, a 5% reduction in community transmission due to other NPIs would avert about 59% of the projected mortality (figure 4b, magenta curve). A 20% reduction in community transmission ($c_r = 0.2$) would avert about 60% of the projected mortality (figure 4b, green curve). See table 3 for a summary of the results. Similar results are obtained with respect to daily mortality (figure S2 in the electronic supplementary material).

### 2.3.3. Assessing the combined impact of different kinds of masks, mask compliance and other non-pharmaceutical interventions

Additional simulations are carried out to assess the combined impact of using different kinds of face masks (cloth masks with an estimated efficacy of 30%, improved cloth masks with an estimated efficacy of 50%, surgical masks with an estimated efficacy of 70%, and respirators or equivalent with efficacy of approx. 95%) and other NPIs on the cumulative COVID-19 mortality. For these simulations, we choose 30 April 2021 as the reference time for comparison purposes. The results obtained, depicted in figure 5, show that respirators are far more effective in curtailing COVID-19 mortality than using any of the other mask types, as expected. Furthermore, the effectiveness of each mask type is enhanced if the face masks intervention is combined with increase in baseline levels of other NPIs. For instance, if Americans use cloth masks only with a compliance of 50%, from 12 October 2020, 750 000 cumulative mortality will be recorded by 30 April 2021 (figure 5a, magenta curve). This represents a 32% increase in the cumulative mortality recorded under the baseline scenario (570 000), where the face masks compliance ($c_m$) and increase in other NPIs ($c_r$) are maintained at their baseline levels (figure 5a, blue curve). On the other

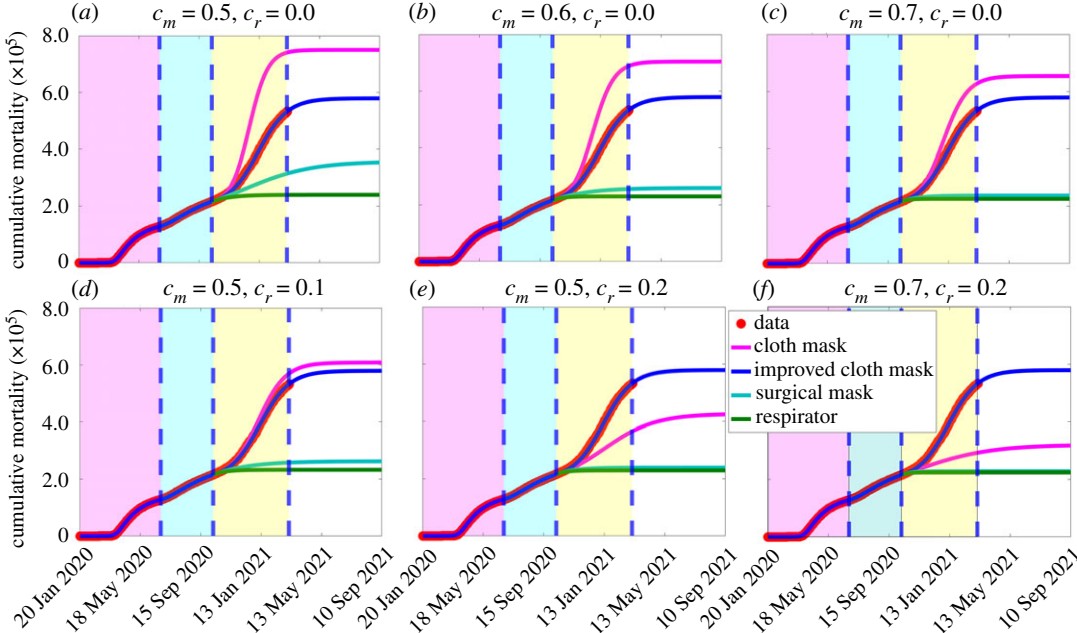

**Figure 5.** Simulations of the basic model (2.1) to assess the impact of masks type (i.e. cloth mask only, improved cloth masks only, surgical masks only, and respirators only), as a function of time, for various levels of face masks compliance ($c_m$) and increasing levels of other NPIs from their baseline ($c_r$), starting 12 October 2021. (a) various mask types with compliance set at 50% (i.e. $c_m = 0.5$) and other NPIs maintained at baseline (i.e. $c_r = 0$). (b) various mask types with compliance set at 60% (i.e. $c_m = 0.6$) and other NPIs maintained at baseline (i.e. $c_r = 0$). (c) various mask types with compliance set at 70% (i.e. $c_m = 0.6$) and other NPIs maintained at baseline (i.e. $c_r = 0$). (d) various mask types with compliance set at 50% (i.e. $c_m = 0.5$) and other NPIs increased to 10% from their baseline (i.e. $c_r = 0.1$). (e) various mask types with compliance set at 50% (i.e. $c_m = 0.5$) and other NPIs increased to 20% from their baseline (i.e. $c_r = 0.2$). (f) various face mask types with compliance set at 70% (i.e. $c_r = 0.7$) and other NPIs increased to 20% from their baseline (i.e. $c_r = 0.2$). Red dots represent observed cumulative mortality data. The other parameter values used in the simulations are as given in table 1 and the third wave parameter values in table 2. The shaded magenta and green regions represent the beginning and end of the first and second pandemic waves in the USA, respectively. The shaded yellow region represents the beginning of the third pandemic wave until 11 March 2021. Dashed vertical blue lines demarcate the three waves.

hand, if surgical masks are used from 12 October 2020, the cumulative mortality to be recorded on 30 April 2021 will be 335 000 (figure 5a, cyan curve), which represents a 41% decrease from the baseline scenario (depicted in figure 5a, blue curve). Furthermore, if respirators are used from 12 October 2020, the cumulative mortality to be recorded on 30 April 2021 will be 240 000 (figure 5a, green curve), which represents a 58% decrease from the baseline scenario (depicted in figure 5a, blue curve). Improved cloth masks correspond to the baseline scenario (figure 5a, blue curve). If the face mask compliance ($c_m$) is increased by 20% (i.e. if $c_m$ is now set to 0.7, instead of 0.5 above), then using cloth masks only from 12 October 2020 will result in a 12% increase in cumulative mortality by 30 April 2021 (figure 5c, magenta curve). For this scenario, surgical masks will reduce the baseline cumulative mortality by 58% (figure 5c, cyan curve), while respirators reduce the mortality by 60% (figure 5c, green curve).

We further simulated the scenario where face mask compliance is kept at baseline (i.e. $c_m = 0.5$) and NPIs increased by 10% and 20% from their baseline (i.e. $c_r = 0.1$ and $c_r = 0.2$), starting from 12 October 2020. In this scenario, cloth masks increase baseline cumulative mortality recorded by 30 April 2021 by 6%, when $c_r = 0.1$ (figure 5d, magenta) and reduce cumulative mortality by 30% when $c_r = 0.2$ (figure 5e, magenta). When $c_r = 0.1$, surgical masks (figure 5d, cyan curve) and respirators (figure 5d, green curve) decrease the cumulative mortality by 54% and 59%, respectively. These percentage reductions due to surgical masks and respirators use increase to 58% (figure 5e, cyan curve) and 60% (figure 5e, green curve), respectively, if $c_r = 0.2$. Finally, if face masks compliance is increased to 70% (i.e. $c_m = 0.7$), while NPIs remain at 20% above their baseline levels (i.e. $c_r = 0.2$), the use of cloth mask can reduce the baseline cumulative mortality recorded on 30 April 2021 by 46% (figure 5f, magenta curve). Here, surgical masks (figure 5f, cyan curve) and respirators (figure 5f, green curve) reduce the baseline cumulative mortality recorded by 30 April 2021 by 60% and 61%, respectively.

### 2.3.4. Assessment of the impact of vaccination and waning natural and vaccine-derived immunity

Three highly effective vaccines (developed by Moderna Inc., Pfizer Inc. and Janssen Biotech Inc.) were given emergency use authorization (EUA) by the United States Food and Drug Administration [47–51]). While the Pfizer and Moderna vaccines received EUA, and began to be administered in the USA in December 2020, the Johnson & Johnson vaccine (developed by Janssen Biotech Inc.) received EUA in March of 2021. The Pfizer and Moderna vaccines, administered in two doses, have estimated protective efficacy of about 95%, while the single-dose Johnson & Johnson vaccine has estimated efficacy of 70% [47,48,52–56].

It is known that the immunity derived from these vaccines, as well as natural immunity derived from recovery from prior COVID-19 infection, wanes over time [57–59]. In particular, it is estimated that the duration of natural immunity is approximately nine months, while that of vaccine-derived immunity is approximately six months [57,58]. It is, therefore, instructive to assess the potential impact of the loss of vaccine-derived and natural immunity on the trajectory of the pandemic. To achieve this, the basic model (2.1) will be slightly modified to enable for the assessment of these factors, as described below.

### 2.3.4. I. Assessment of the impact of waning natural immunity

In order to account for waning natural immunity, we consider a slightly modified version of the basic model (2.1), where a certain proportion of recovered individuals lose their infection-acquired immunity and revert to the susceptible class at a rate $\rho$ (i.e. $1/\rho$ is the average duration of natural immunity to COVID-19). This modified version of the basic model is then simulated, using the parameter values given in tables 1 and 2 with $1/\rho$ set to six months (in line with [57,58]) and the four mask types considered in this study (namely cloth masks, improved cloth masks, surgical masks and respirators). These simulations are also carried out for varying levels of increases in the baseline values of other NPIs (as measured by the parameter $c_r$). The simulation results obtained are depicted in figure 6. The results in this figure show, for the case where NPIs are maintained at their baseline values at the beginning of the third wave (i.e. $c_r = 0$) a marked increase in the daily COVID-19 mortality, regardless of mask type, in comparison with the baseline case where immunity does not wane (figure 6, blue curve). The size of the increase in disease burden increases with decreasing efficacy of the mask type used. For instance, for this setting with waning natural immunity, while the use of cloth and improved cloth masks result in significant increase in daily mortality even with moderate increases in the baseline levels of other NPIs (figure 6a,b), using surgical masks or respirators would eliminate the likelihood of a significant outbreak of the pandemic if combined with

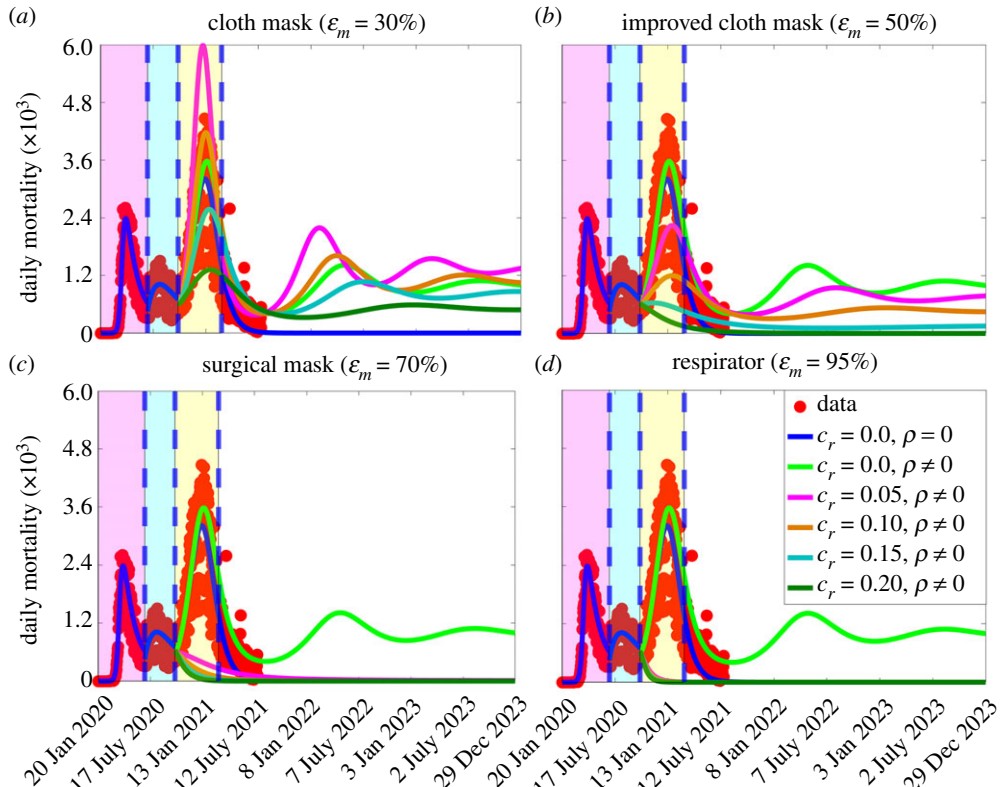

**Figure 6.** Simulations of the basic model (2.1) to assess the combined impact of masks (by type), waning natural immunity and reduction in community transmission due to other NPIs (implemented at the beginning of the third wave) on the daily COVID-19 mortality in the USA. (*a*) Only cloth masks are used. (*b*) Only improved cloth masks are used. (*c*) Only surgical masks are used. (*d*) Only respirators (or equivalent) are used. For the blue and light green curves (baseline scenarios) in figure (*a–d*), the mask efficacy is maintained at 50% (i.e. $\varepsilon_m = 0.5$) and natural immunity does not wane (blue curve: $\rho = 0$) or wanes (light green curve: $\rho \neq 0$). It is assumed that the average duration of immunity is six months, which corresponds to $\rho = 5.6 \times 10^3 \, \text{day}^{-1}$. The other parameter values used in the simulations are given in tables 1 and 2. The shaded magenta and green regions represent the beginning and end of the first and second pandemic waves in the USA, respectively. The shaded yellow region represents the beginning of the third pandemic wave until 11 March 2021. Dashed vertical blue lines demarcate the first three waves.

even a very low increase in the baseline level of other NPIs implemented at the beginning of the third wave (figure 6*c,d*), such as $c_r = 0.05$.

Furthermore, it is evident from the simulations in figure 6 that waning natural immunity could trigger subsequent waves of the COVID-19 pandemic in the USA, and the number, severity, timing of the peak, and duration of the subsequent waves depend on the type of mask used and the level of increases in the coverage of the baseline values of other NPIs ($c_r$). For example, if cloth and improved cloth masks are used, the USA may experience a fourth wave of the COVID-19 pandemic around early May of 2022. The predicted new waves or outbreaks of the pandemic can be suppressed (or eliminated) if surgical masks and respirators are used in combination with increases in the coverage of other NPIs, in comparison with their baseline values at the beginning of the third wave (figure 6*c,d*). Furthermore, the time-to-elimination of the pandemic is shorter when respirators are used, compared with when surgical masks are used. In summary, the simulations depicted in figure 6 show that waning natural immunity causes an increase in the pandemic burden. Furthermore, the effective control of the pandemic, under this scenario of waning natural immunity, necessitates the use of high-quality masks (notably surgical masks or respirators), particularly if the high-quality mask-based strategy is combined with a strategy that increases the values of other NPIs from their baseline levels at the beginning of the third wave.

### 2.3.4. II. Assessment of the impact of combined vaccination and waning vaccine-derived and natural immunity

The basic model can be adapted to include the use of a vaccine, by moving a certain proportion of the susceptible individuals into the compartment of recovered individuals, at a rate $\xi$. That is, we consider

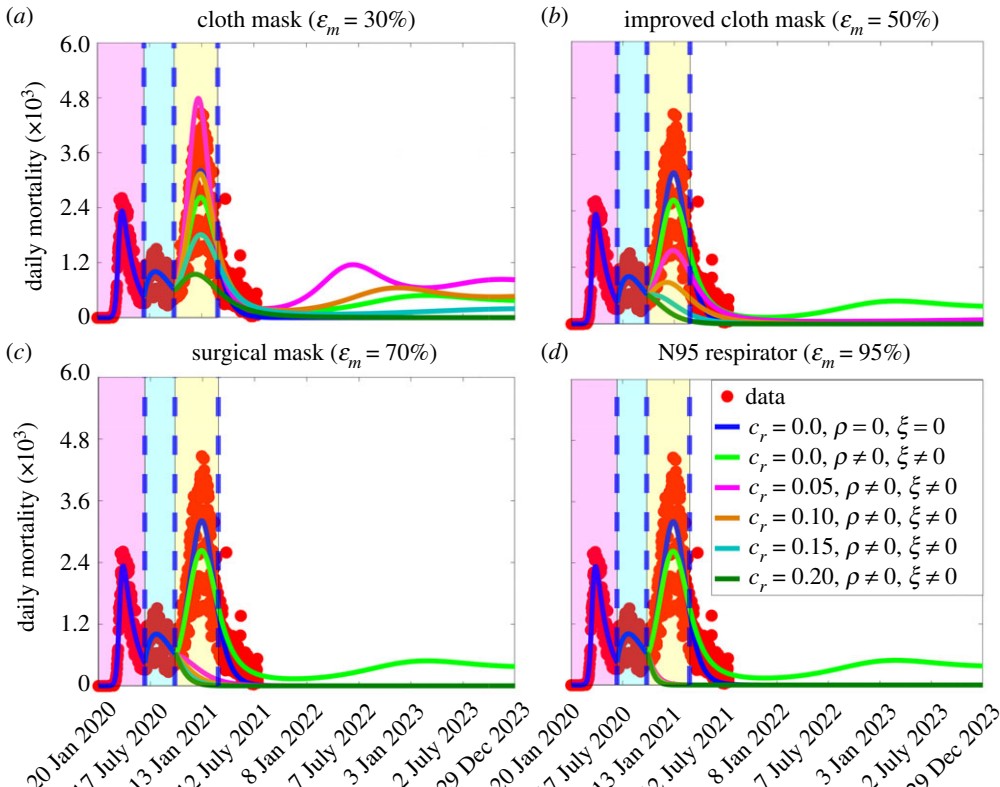

**Figure 7.** Simulations of the basic model (2.1) to assess the combined impact of masks, vaccination, waning natural and vaccine-derived immunity and reduction in community transmission due to other NPIs on the daily COVID-19 mortality as a function of time, and various mask types, for the USA. Reduction in community transmission due to other NPIs and vaccination are implemented at the beginning of the third wave (i.e. from 12 October 2020). For these simulations, only the Pfizer vaccine is used (with estimated efficacy of 95%) and the vaccination rate is set at $\xi = 7.4 \times 10^4 \, \text{day}^{-1}$ (i.e. 250 000 individuals are fully vaccinated daily). The vaccine-derived and natural immunity wane after six months (which corresponds to $\rho = 5.6 \times 10^3 \, \text{day}^{-1}$). (a) Only cloth masks are used. (b) Only improved cloth masks are used. (c) Only surgical masks are used. (d) Only respirators (or equivalent) are used. For the blue and light green curves (baseline scenarios) in figures (a–d), the mask efficacy is maintained at 50% (i.e. $\varepsilon_m = 0.5$). The other parameter values used in the simulations are given in tables 1 and 2. The shaded magenta and green regions represent the beginning and end of the first and second pandemic waves in the USA, respectively. The shaded yellow region represents the beginning of the third pandemic wave until 11 March 2021. Dashed vertical blue lines demarcate the three waves.

the scenario where fully vaccinated susceptible are immunized against infection (and remain immune, in the recovered class, until their vaccine-derived immunity wanes). The slightly modified version of the model will now be simulated to assess the combined impact of waning vaccine-derived and natural immunity. For illustrative purposes, we consider the scenario where only the Pfizer vaccine is used in the population (so that we set $\varepsilon_v = 0.95$ in the simulations of the modified model). Furthermore, we consider the scenario that the vaccine was available at the beginning of the third wave of the pandemic (12 October 2020) and that an average of 250 000 individuals are fully vaccinated daily. This corresponds to a vaccination rate of $\xi \approx 7.4 \times 10^{-4} \, \text{day}^{-1}$. We also assume that both the vaccine-derived and natural immunity wane after six months (i.e. $\rho = 5.6 \times 10^3 \, \text{day}^{-1}$).

The simulation results obtained, depicted in figure 7, show that the waning of vaccine-derived and natural immunity would cause an increase in disease burden (triggering a fourth wave of the pandemic in the USA that would peak around February 2023), regardless of the type of mask used, if the coverage level of other NPIs is not increased from its baseline value at the beginning of the third wave (figure 7; light green curves). The increase in disease burden is significantly lower than the increase reported in figure 6, where vaccination was not used (i.e. vaccination, even at the low daily vaccination rate of 250 000 day$^{-1}$, greatly reduces the burden of the pandemic even if both the vaccine-derived and natural immunity wane after a period of six months). This figure further shows that even the use of improved quality masks can lead to the elimination of the

pandemic if the vaccination programme is combined with a strategy that increases the baseline values of other NPIs, in comparison with the baseline (such as $c_r > 0.05$). We also simulated the scenario where the vaccination programme is implemented, but both the vaccine-derived and natural immunity do not wane. The results obtained are depicted in figure S3 of the electronic supplementary material. This figure shows a much lower disease burden and shorter time-to-elimination, in comparison with the case (depicted in figure 7), where the vaccine-derived and natural immunity wane (after six months).

# 3. The extended model: effect of behaviour change with respect to mask use

## 3.1. Formulation of extended model

Although the basic model (2.1) provides insights on the population-level impact of various types of face masks implemented as a sole intervention, or in combination with other NPIs, it does not allow for the assessment of the impact of changes in face mask adherence. For example, upon taking the oath of office in the middle of the third wave on 20 January 2021, President Biden issued a mask mandate in all Federal buildings and public transportation systems [60,61]. To account for changes in face mask usage, the basic model (2.1) is now extended and reformulated into a multi-group structure, which entails stratifying the total population into three groups: those who do not wear masks (Group 1); those who habitually wear other types of masks, except respirators (or equivalent), such as cloth masks, improved cloth/poorly fitted surgical masks, or properly fitted surgical masks (Group 2); and those who habitually wear respirator or equivalent (Group 3) only in public. The resulting extended multi-group model also allows for back-and-forth transitions between the three groups.

To develop the extended model, the total population at time $t$, denoted by $N(t)$, is split into the total population of individuals who do not wear face masks in public (denoted by $N_1(t)$), the total population of individuals who habitually wear other types of masks except respirators (or equivalent) only (denoted by $N_2(t)$) and the total population of individuals who habitually wear the respirator (or equivalent) only (denoted by $N_3(t)$). Each of these subgroups is further subdivided into seven mutually exclusive epidemiological compartments of susceptible ($S$), exposed/latent ($E$), pre-symptomatic infectious ($P$), symptomatic infectious ($I$), asymptomatic infectious ($A$), hospitalized ($H$) and recovered ($R$) compartments, so that

$$N_i(t) = S_i(t) + E_i(t) + P_i(t) + I_i(t) + A_i(t) + H_i(t) + R_i(t); \quad i = 1, 2, 3.$$

The extended multi-group model also allows for the assessment of the impact of other NPIs implemented during the beginning of the third wave of the pandemic (as represented by the parameter $c_r$, discussed earlier). The multi-group extended model is formulated via the same approach used in the formulation of the (single group) basic model (2.1). The equations for the rate of change of the state variables of the extended model are given by equations (S.1)–(S.3) in the electronic supplementary material. Parameters that represent change in face mask usage (i.e. the back-and-forth transitions between groups) are denoted by $\alpha_{ij}$; $i$, $j = 1, 2, 3$ with $i \neq j$, as presented in table S2 of the electronic supplementary material. Furthermore, the state variables and parameters of the extended model are tabulated in electronic supplementary material, tables S3–S4, respectively). The remaining parameters of the extended model are the same as those used for the basic model (except for the indices, indicating the corresponding group the parameter is defined for). The extended model is also fitted using the cumulative mortality data for the USA, in the electronic supplementary material.

## 3.2. Results from the extended model

The extended model, given by equations (S.1)–(S.3) in electronic supplementary material, is now simulated to assess the population-level impact of change of masking behaviour (measured in terms of the parameters $\alpha_{ij}$, $i$, $j = 1, 2, 3$ with $i \neq j$) on the transmission dynamics and control of the disease, for various increases in the baseline coverage levels of other NPIs ($c_r$) implemented after 12 October 2020. The results obtained, depicted in figure 8, show that increases in the rate of behaviour change of non-wearers to adopt to habitually wear respirators ($\alpha_{13}$) is more effective, in curtailing peak daily mortality and peak daily hospitalization, than corresponding increases in the rate at which non-wearers decide to wear surgical masks ($\alpha_{12}$). For instance, if 500 000 non-wearers change their behaviour and begin to wear surgical

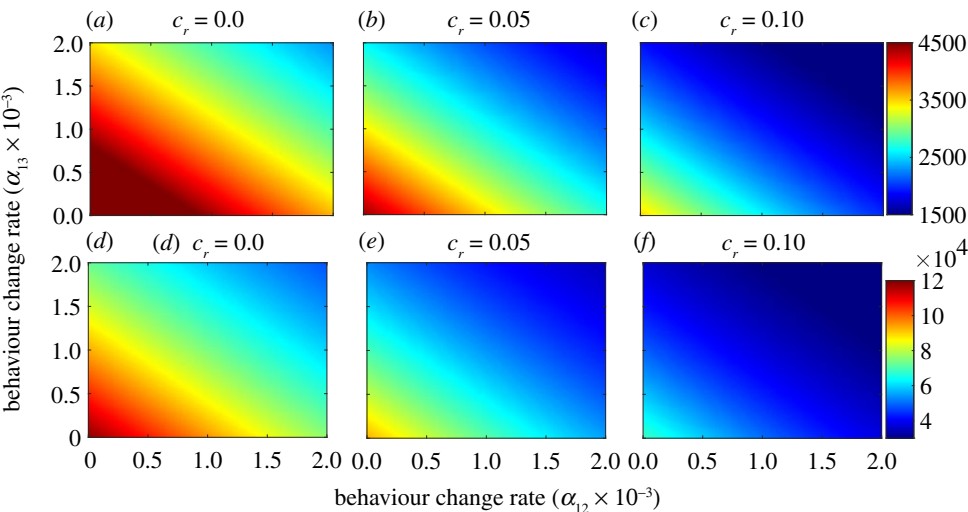

**Figure 8.** Heatmaps of the extended model (S.1)–(S.3), given in electronic supplementary material, showing the peak daily mortality (*a–c*) and peak daily hospitalizations (*d–f*), as a function of the daily rate at which non-wearers opt to wear surgical masks ($\alpha_{12}$) or respirators ($\alpha_{13}$) for increasing baseline coverage levels of other NPIs ($c_r$), starting 12 October 2020. (*a*) $c_r = 0.0$, (*b*) $c_r = 0.05$, (*c*) $c_r = 0.1$, (*d*) $c_r = 0.0$, (*e*) $c_r = 0.05$ and (*f*) $c_r = 0.1$. The values of the other parameters of the extended model used in these simulations are as given in table 1 and electronic supplementary material, table S6.

masks only (i.e. $\alpha_{12} = 0.0015$ day$^{-1}$, and $\alpha_{13} = 0$), the baseline maximum peak daily mortality obtained when wearers do not abandon mask wearing from 12 October 2020 onward (5316 deaths, obtained from figure 8*a*) reduces to 3961 (which represents a 25.5% reduction in the baseline maximum peak daily mortality). However, if the 500 000 non-wearers opt to wear respirators or equivalents, instead (so that, $\alpha_{13} = 0.0015$ and $\alpha_{12} = 0$), the baseline maximum peak daily mortality reduces to 3862 (corresponding to 27.4% reduction in the baseline maximum peak daily mortality). On the other hand, if the 500 000 non-wearers choose to split 50–50 between the two masks (i.e. half wear surgical masks only and the remaining half wear respirators only), the baseline maximum peak daily mortality reduces to 3920 (which represents a 26.3% reduction in the baseline maximum peak daily mortality). It should be mentioned that lower reductions in peak daily mortality and hospitalizations are obtained if the comparison is between respirators and cloth or improved cloth masks, or between surgical masks and cloth or improved cloth masks.

Figure 8 also shows that greater reductions in the peak daily mortality and the peak daily hospitalizations are achieved if the face masks use strategy is complemented with increasing baseline coverages of other NPIs (figure 8*c,f*, in comparison with figure 8*a,b,d* and *e*). For instance, for the above case where 500 000 non-wearers change their behaviour and opt to wear one of the two masks types, combined with a 10% increase in baseline coverage of other NPIs ($c_r = 0.1$), the maximum baseline daily mortality (5316) reduces to 2257 and 2183, respectively, if the non-wearers choose to wear surgical masks or respirators exclusively (figure 8*c*). This represents a 57.5% and 58.9% reduction in the baseline maximum peak mortality, respectively (recall that the corresponding reductions, under this scenario but with $c_r = 0$, described above, were, respectively, 25.5% and 27.4%). Additionally, if the 500 000 non-wearers are split equally between new surgical masks and respirator wearers, then the maximum baseline daily mortality (5316) reduces to 2233 (representing a 58% reduction in the baseline maximum peak mortality). Similar results are shown with respect to the baseline maximum daily hospitalization in figure 8*d–f*.

We further simulated the extended model to compare the population-level effectiveness of the four mask types (cloth, improved cloth or poorly fitted surgical, properly fitted surgical and respirators), for the case where all mask wearers in the community choose to wear only one mask type, for various initial sizes of the total population of mask wearers (denoted by $N_2(0) + N_3(0)$). The results are depicted in figure 9. This figure shows a devastating level of cumulative mortality (about 1.15 million deaths by 30 April 2021) under the worst-case scenario, where no non-wearer in the community begins to wear a face mask in public from the beginning of the third wave, 12 October 2020 (figure 9*a–c*, purple curves). Similar devastating numbers are recorded for the daily mortality (with a peak daily mortality of about 9400 on 4 January 2021) under this scenario (figure 9*d–f*). For the case when 65% of the population were wearing face masks at the beginning of the third wave of the pandemic

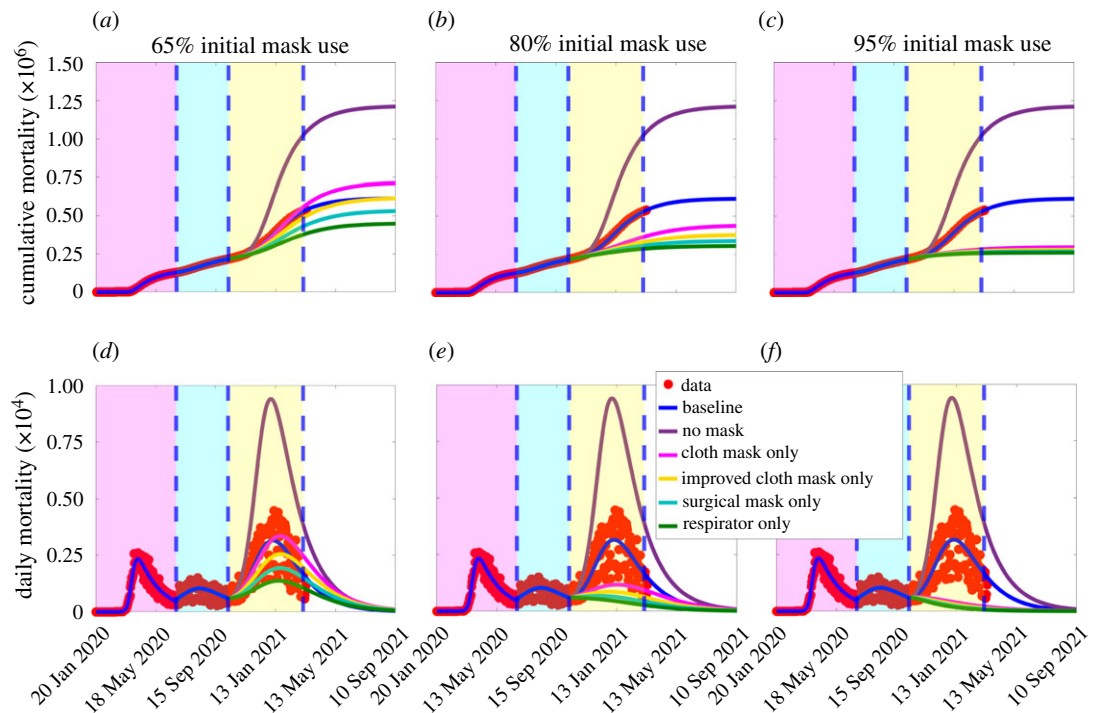

**Figure 9.** Simulations of the extended model (S.1)–(S.3), given in electronic supplementary material, showing the effects of mask types and initial size of total population of mask wearers ($N_2(0) + N_3(0)$) on the cumulative (*a*–*c*) and daily (*d*–*f*) COVID-19 mortality. (*a*,*d*): total initial size of mask wearers is 65%; (*b*,*e*) total initial size of mask wearers is 80%; (*c*,*f*): total initial size of mask wearers is 95%. The values of the other parameters of the extended model used in the simulations are as given in table 1 and electronic supplementary material, tables S5 and S6. The shaded magenta and green regions represent the beginning and end of the first and second pandemic waves in the USA, respectively. The shaded yellow region represents the beginning of the third pandemic wave until 11 March 2021. Dashed vertical blue lines demarcate the three waves.

(i.e. $N_2(0) + N_3(0) = 65\%$ of the population), the cumulative mortality recorded on 30 April 2021 under the baseline scenario (i.e. using the baseline parameter values in table 1 and electronic supplementary material, tables S5 and S6) is 585 000 (figure 9*a*, blue curve). This represents a 49% reduction in cumulative mortality under the worst-case scenario.

If all mask wearers in the community opt for cloth masks only, from the beginning of the third wave of the pandemic (i.e. from 12 October 2020), the cumulative mortality recorded by 30 April 2021 is 650 000 figure 9*a*, magenta curve). Although this represents a 43% reduction in the worst-case cumulative mortality, it is higher than the cumulative mortality recorded under the baseline scenario. Using improved cloth or improperly fitted surgical masks only, on the other hand, results in cumulative mortality of 561 000 by 30 April 2021 (figure 9*a*, yellow curve), representing a 51% decrease from the worst-case cumulative mortality and a 4% decrease from the baseline cumulative mortality. For this scenario, there is a threshold time (around early September 2021) and cumulative mortality (about 612 000) after which the recorded cumulative mortality with improved cloth mask exceeds the baseline cumulative mortality. (compare blue and yellow curves in figure 9*a*). Similarly, using properly fitted surgical masks only results in cumulative mortality of 488 000 by 30 April 2021 (figure 9*a*, cyan curve), representing a 58% decrease from the cumulative mortality recorded under the worst-case scenario and a 17% decrease from the baseline. However, if respirators are used, the projected cumulative mortality by 30 April 2021 is 418 000 (figure 9*a*, green curve), corresponding to a 64% decrease from the worst-case cumulative mortality scenario and a 29% decrease from the baseline cumulative mortality. Similar results are obtained with respect to the daily mortality (figure 9*d*).

Figure 9 further shows a marked decrease in cumulative mortality with increasing initial size of the population of individuals who opt to wear face masks from the beginning of the third wave of the pandemic (figure 9*b*). For instance, if the initial size of mask wearers is increased from 65% to 80%, the reductions in worst-case cumulative mortality are much higher than the reductions achieved when the initial population of mask wearers was 65% (compare (*a*) with (*b*)). Furthermore, unlike in the case where the initial population of mask wearers is 65%, the cumulative mortality recorded for all four

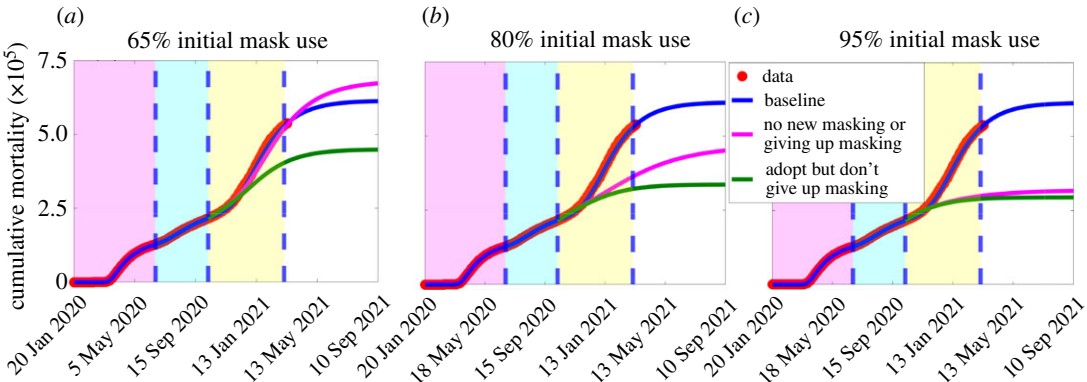

**Figure 10.** Simulations of the extended model (S.1)–(S.3), given in electronic supplementary material. Assessment of the impact of increases in the total initial size of mask wearers on cumulative mortality. (*a*) The initial population of mask wearers is 65%. (*b*) The initial population of mask wearers is 80%. (*c*) The initial population of mask wearers is 95%. The magenta curve (no behaviour change) depicts the scenario in which there is no new masking and already masking individuals do not give up masking, while the green curve depicts the scenario in which new people opt for masking and existing masking individuals do not abandon masking. The magenta, cyan and yellow regions denote the first pandemic wave, the second pandemic wave, and the portion of the third pandemic wave up to 11 March 2021, respectively, while the dashed vertical blue lines demarcate the first and second wave and the portion of the third wave up to 11 March 2021. The values of the other parameters of the extended model are as given in table 1 and electronic supplementary material, tables S5 and S6.

mask types when the initial size of wearers is 80% is lower than the baseline cumulative mortality (figure 9*b*), with respirators much more effective than surgical and cloth masks. When 95% of members of the community chooses to wear a mask initially (i.e. from the very beginning of the third wave of the pandemic), our simulations show a dramatic decrease in cumulative mortality, in comparison with the baseline, regardless of the mask type used (figure 9*c*). Although the mask effectiveness is essentially the same, regardless of the mask type chosen, respirators are marginally better than the others, while properly fitted surgical masks are also marginally better than cloth masks (figure 9*c*). Figure 9*f* further shows that if 95% of the US populace opted to wear respirators in public, from the beginning of the third pandemic wave, daily COVID-19 mortality recorded in the country will fall below one by mid-June 2021. The same dramatic reduction in daily mortality will be achieved (under this scenario), but a month later, if surgical masks are preferred instead. However, if basic and improved cloth masks are preferred, such reduction can only be achieved around mid-January 2022 and December 2021, respectively. In summary, the simulations in figure 9*a–c* show that respirators are always more effective in reducing cumulative mortality (in comparison with either the worst-case or baseline scenario), than any of the other mask types considered in this study. Furthermore, these results show that the population-level effectiveness of the four mask types is essentially identical if the initial size of mask wearers is very high (e.g. greater than 85%). The results show a similar pattern with respect to daily mortality (figure 9*d–f*).

The extended model is further simulated to assess the combined impact of the cumulative initial number of individuals who wear face masks and masking behaviour on the spread of the pandemic. For these simulations, we take the reference point to be 12 October 2020 (i.e. the beginning of the third wave of the pandemic). Figure 10 shows that, for the case where 65% of Americans are wearing face masks by 12 October 2020, the projected cumulative mortality by 30 April 2021 for the scenario with no new mask wearers ((i.e. $\alpha_{12} = \alpha_{13} = 0$) and existing mask wearers do not abandon mask wearing (i.e. $\alpha_{21} = \alpha_{31} = 0$) exceeds the projected cumulative mortality for the baseline scenario (figure 10*a*, gold curve; in comparison with blue curve) by about 4%. If, on the other hand, more Americans choose to wear masks after 12 October 2020 (in addition to the 65% initially) no new wearers abandon mask-wearing (i.e. $\alpha_{12}$ and $\alpha_{13}$ are kept at their baseline values, while $\alpha_{21}$ and $\alpha_{31}$ are set to zero), the projected cumulative mortality is decreased below the projected cumulative mortality at baseline, by about 26% (figure 10*a*, green curve; in comparison with blue curve). For the case where the initial percentage of Americans who were wearing face masks by 12 October 2020 is 80%, our simulations show that even the case with no new mask wearers resulted in a decrease in projected baseline cumulative mortality, by about 31% when masks wearers do not stop wearing masks (figure 10*b*, magenta curve; in comparison with blue curve). This reduction increases to 43% if more Americans choose to wear masks, and no mask wearers abandon their mask-wearing behaviour

(figure 10*b*, green curve, in comparison with blue curve). Further reductions in projected cumulative mortality are recorded if the percentage of initial mask wearers by 12 October 2020 is 95% (figure 10*c*).

The simulations in figure 10 clearly show that there is a threshold level of initial size of population of mask wearers above which the projected baseline cumulative mortality by 30 April 2021 will always be reduced even in the absence of new wearers ($\alpha_{12} = \alpha_{13} = 0$), while existing wearers remain wearers (i.e. $\alpha_{21} = \alpha_{31} = 0$, and $\alpha_{23}$ and $\alpha_{32}$ maintained at baseline). Larger reductions in the projected baseline cumulative mortality are recorded if more new wearers are attracted (and existing wearers do not opt out of wearing masks), as expected (green curves in figure 10). Similar results for different levels of other NPIs, as opposed to different initial sizes of mask wearers (at the beginning of the third wave of the pandemic), are presented in electronic supplementary material, figure S4.

Here, too, the extended model (S.1)–(S.3), given in electronic supplementary material, is slightly modified (using the approach described in §2.3.4 for the basic model) to account for the impact of vaccination and waning vaccine-derived and natural immunity. The numerical simulation results obtained, depicted in figures S6 and S7 of the electronic supplementary material, show similar pandemic trajectory as those reported in §2.3.4 for the basic model. That is, in the absence of vaccination, waning (natural and vaccine-derived) immunity increases the disease burden. In the presence of vaccination (even with the low rate of 250 000 vaccine doses delivered per day used in the numerical simulations of the extended model) waning of immunity still increases the disease burden but at a lower rate than the burden generated without vaccination.

# 4. Discussion and conclusion

With numerous classes of respiratory protection available, including cloth masks, surgical/procedure masks and respirators, there has been little clarity on the widespread use of respirators among the general public and to what extent their use might curtail a pandemic driven by the respiratory transmission route. In this study, we formulated a basic dynamic epidemiology model for assessing the population-level impact of different types of face masks and other NPIs on curtailing and mitigating the burden of the COVID-19 pandemic in the USA. The basic model, which takes the form of a deterministic system of nonlinear differential equations, is calibrated based on updated COVID-19 mortality data from the three waves of the COVID-19 pandemic in the USA.

During the early stages of the COVID-19 pandemic in the USA, masks were in limited supply. In particular, respirators were in exceptionally limited supply, and prioritized for frontline healthcare workers. We carried out numerical simulations of the basic model to assess the hypothetical scenario where masks are in abundant supply, and are deployed to the general public during the early stages of the pandemic. The simulations show that the early implementation of this masking strategy, particularly if respirators or well-fitted surgical masks are universally adopted nationwide, would have greatly suppressed the burden (as measured in terms of cumulative mortality) and potential of the COVID-19 pandemic in the USA. In particular, there would have been no significant outbreak of the pandemic in the USA if four out of every five Americans had chosen to mask up early and prioritized respirators or well-fitting surgical masks. Our study shows that the prospect of COVID-19 elimination in the USA is greatly enhanced if a sizable proportion of the US population opt to wear high-quality masks, notably respirators or tightly fitted surgical/procedure masks, from the onset of the pandemic (at least 40% if respirators are used; and 55% if surgical/procedure masks are used). The compliance level needed to achieve such elimination is significantly higher (about 75%) if data for the first wave of the pandemic is used to calibrate the model, instead of the data for the third wave (see figure S1 in electronic supplementary material; this is because the baseline value of the control reproduction number of the model, $\mathbb{R}_c \approx 3.1$, is much higher during the first wave, in comparison with its value during the third wave). In other words, our study supports the 'hit hard, hit early' public health containment strategy, but with high-quality masks. Thus, this study suggests that high-efficacy respirator protection should be stockpiled and made readily available to members of the public in times of future major outbreaks of respiratory diseases. This result is consistent with reports in [62–64], which emphasize the superiority of mask-based strategies using high-quality masks. The mask compliance needed to eliminate the pandemic using respirators drops to almost one in five Americans if the masking strategy is complemented with other NPIs that can reduce community transmission by 20% from their baseline at the beginning of the third wave of the pandemic. This result is consistent with that reported by Hellewell *et al.* [65], which suggests that even minor

decreases in community transmission of COVID-19 can have a significant impact on efforts aimed at containing the pandemic.

Our study shows that communities that prioritize the use of low-efficacy face coverings, such as cloth masks, must complement this strategy with other community transmission-reducing NPIs (such as social distancing, self-isolation, or even community lockdowns) at very high levels of effectiveness and coverage in order to effectively control the spread of the pandemic. On the other hand, communities that adopt high-efficacy respiratory protection would only need to, in the case of well-fitted surgical masks, complement the masking strategy with other NPIs at low to moderate levels of effectiveness and coverage. No complementing with other NPIs would be needed if respirators or equivalent, at moderate coverage, were used. Prioritizing highly efficient respiratory protection, particularly respirators, if adopted by 95% of the populace at the beginning of the third pandemic wave in the USA, is estimated to eliminate the daily COVID-19 mortality by mid-June, 2021.

We showed that if the overall face masks compliance is moderate (e.g. 50% compliance), prioritizing low-efficacy face masks will not be effective in reducing the burden of the pandemic. In fact, the use of low-efficacy masks in this setting (with 50% compliance) increases the disease burden, in comparison with the baseline scenario. Opting, instead, for well-fitting surgical masks or respirators (in this 50% compliance scenario) will lead to a significant reduction in pandemic burden. If masks use compliance in the community is moderate (e.g. 70%), our study shows that surgical masks and respirators are almost equally as effective in reducing pandemic burden, with respirators being marginally more effective. However, the effectiveness of the face mask strategy, particularly if masks other than respirator or equivalent are prioritized, is greatly enhanced if the mask-based strategy is combined with other NPIs that reduce community transmission from its baseline level (e.g. NPIs that reduce baseline community transmission by at least 20%). Our simulations clearly support the supremacy of respirators over all other mask types in reducing and mitigating the burden of the COVID-19 pandemic in the USA.

Recent studies have shown that immunity against COVID-19, derived either through vaccination or from recovery from prior infection with COVID-19, wanes with time (on a timescale of six to nine months). We slightly modified our basic model to allow for the assessment of the combined population-level impact of a vaccine against COVID-19 and waning vaccine-derived and natural immunity. We demonstrated, through numerical simulation of the slightly modified basic model, that waning vaccine-derived and natural immunity could trigger new waves of the COVID-19 pandemic in the USA, and the number, severity and duration of the new pandemic waves depend on the quality of mask type used and the level of increase in the coverage of other NPIs (in comparison with their baseline values during the onset of the third wave of the pandemic in the USA). Furthermore, we showed that the burden of the pandemic (as measured in terms of daily mortality) increases with increasing waning rate of the natural and vaccine-derived immunity. When these (natural and vaccine-derived) immunity wane over time, our study showed that the effective control of the COVID-19 pandemic will necessitate the use of high-quality masks, such as surgical masks or respirators. The prospect of this effective control is enhanced if the masks-based strategy (using the high-quality surgical masks or respirators) is complemented with a strategy that increases the coverage of other NPIs (in relation to their baseline values during the onset of the third wave of the pandemic in the USA). Additionally, we demonstrated that vaccination (using the Pfizer or Moderna vaccines), even when administered at relatively low daily rates (such as vaccinating 250 000 individuals every day, starting from the onset of the third wave), leads to a significant reduction in the burden of the pandemic (compared with the case with no vaccination). This reduction is more pronounced when vaccine-derived and natural immunity do not wane over time.

In order to assess the impact of human behaviour, with regard to whether or not users choose to wear face masks in public, as well as relaxing mask mandates, we extended the basic model into a three-group model in which the total population is subdivided into the individuals who do not habitually wear face masks in public (Group 1), those who habitually wear cloth or surgical masks (Group 2), and those who habitually wear respirators or equivalent (Group 3). This multi-group structure allows for the assessment of the impact of the initial sizes of the three subgroups and the adoption, adherence to, or relaxation of mass mandates on the burden of the pandemic.

Numerical simulations of the extended multi-group model demonstrate the importance of positive change of behaviour, from non-masking to masking, in significantly reducing the burden (cumulative and daily mortality and hospitalizations) of the pandemic, as expected. This reinforces the urgent need for effective community outreach campaigns to improve mask adoption and opting for higher-efficacy options. This study shows that the effectiveness of one mask type over other types is dependent on the overall mask compliance level in the community. We showed that if the mask

compliance level in the community is very high (e.g. 85%), then the four mask types (cloth, improved cloth, surgical, or respirator) are essentially equally as effective. For communities that can maintain high mask-use compliance, the type of mask chosen or adopted in the community may not matter because, for this high compliance setting, any of the four mask types will be very effective in significantly reducing the pandemic burden. This result is consistent with an observational study reported by Wang *et al.* [66], which indicated that universal household use of face masks was about 79% effective in curtailing the spread of COVID-19. As expected, our simulations also showed that more projected COVID-19-related deaths are averted as more individuals adopt mask wearing and while existing wearers do not opt out of wearing masks. This implies that continued adoption of, and strictly complying with, the face mask use strategy, especially if initial masking coverage in the community is high, is beneficial in effectively reducing and mitigating the burden of the COVID-19 pandemic in the USA. This is particularly significant in most nations where vaccine availability continues to be highly limited.

In summary, our study demonstrates that the prospect for the effective control or elimination of COVID-19 in the USA through modifying the public health masking strategy is promising both singly, if respirators are prioritized, or in combination with other NPIs that reduce baseline community transmission. Our results suggest that the COVID-19 pandemic could have been heavily suppressed or eliminated (i.e. the control reproduction number, $\mathbb{R}_c$, could have been reduced to a value less than one) if only around half of the USA population opted to use respirators from the beginning of the pandemic, even in the absence of needing to complement masking with other NPIs. Such elimination can also be achieved if well-fitted surgical masks were opted for, but with compliance of at least 70% and combined with other NPIs that can reduce baseline community transmission by at least 20%. Our study gives a range of plausible NPI-based scenarios for effectively controlling, and mitigating the burden of, the COVID-19 pandemic in the USA that would not necessarily require major economic disruptions, such as imposing community lockdowns or wide-scale business and school closures. Some of the limitation of our modelling study is the assumption that the population is well-mixed, such that everyone is equally likely to mix with everyone else. Additionally, the impact of human choice (e.g. [67,68]) to adopt or not to adopt a masking strategy on the trajectory of the pandemic is not considered. Furthermore, we assume mask efficacy as being constant. However, when not worn properly (i.e. if not tightly fitted), the actual efficacy of the mask type may not be achieved. Nonetheless, modelling studies can provide insight into the transmission dynamics of the disease and how best to effectively combat its spread and mitigate its burden. As in the case of the prior two deadly coronavirus global outbreaks (SARS and MERS), our study suggests that COVID-19 is controllable using basic public health interventions, particularly using a mask-based strategy that prioritizes high-quality masks, particularly respirators at moderate to high levels of compliance.

Data accessibility. Data and relevant code for this research work are stored in GitHub: https://github.com/CSSEGISandData/COVID-19/tree/master/csse_covid_19_data/csse_covid_19_time_series and have been archived within the Zenodo repository: https://zenodo.org/record/5167597.

Authors' contributions. C.N.N. and A.B.G. designed, analysed and parametrized the dynamical models, in addition to drafting (and editing) the manuscript and supplementary material. C.N.N. fitted the models to data and carried out the simulations. J.R.K. contributed in drafting the Introduction section and suggesting edits for the Abstract and Discussion sections of the manuscript. L.M. contributed in suggesting edits for the Abstract, Introduction and Discussion sections. M.H.B. contributed in drafting the Discussion, in addition to suggesting edits for the rest of the manuscript.

Competing interests. We declare we have no competing interests.

Funding. One of the authors (A.B.G.) acknowledges the support, in part, of the Simons Foundation (award no. 585022) and the National Science Foundation (grant no. DMS-2052363). C.N.N. acknowledges the support of the Simons Foundation (award no. 627346).

Acknowledgements. The authors are grateful to the anonymous reviewers for their constructive comments.

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
