## [Peer Review File · Royal Society Open Science]

Review History

RSOS-210699.R0 (Original submission)

Review form: Reviewer 1

Is the manuscript scientifically sound in its present form?

Yes

Are the interpretations and conclusions justified by the results?

Yes

Is the language acceptable?

Yes

Do you have any ethical concerns with this paper?

No

Have you any concerns about statistical analyses in this paper?

Yes

Recommendation?

Accept with minor revision (please list in comments)

Comments to the Author(s)

The authors assess the impact of face masks on the control and mitigation of the COVID-19 pandemic in the US through dynamic models. The methodology is sound. The main concern is about two related model parameters c_m (the proportion of individuals who wear faces masks in public) and ϵ_m (the protective efficacy of face masks to prevent the transmission of infection to a susceptible individual). Although the values of these parameters are carefully chosen, these values were taken with no solid evidence. Considering the impact of these values on the quantitative results, it is important to provide more evidences/survey data on these values. Otherwise, the results would not be statistically sound.

Review form: Reviewer 2 (Salisu Garba)**Is the manuscript scientifically sound in its present form?**

Yes

Are the interpretations and conclusions justified by the results?

Yes

Is the language acceptable?

Yes

Do you have any ethical concerns with this paper?

No

Have you any concerns about statistical analyses in this paper?

No

Recommendation?

Accept with minor revision (please list in comments)

Comments to the Author(s)

See attached review report (Appendix A).

Decision letter (RSOS-210699.R0)

Dear Dr Ngonghala

The Editors assigned to your paper RSOS-210699 "Assessing the impact of widespread respirator use in curtailing COVID-19 transmission in the United States" have now received comments from reviewers and would like you to revise the paper in accordance with the reviewer comments and

any comments from the Editors. Please note this decision does not guarantee eventual acceptance.

Please submit your revised manuscript and required files (see below) no later than 21 days from today's (ie 16-Jul-2021) date. Note: the ScholarOne system will 'lock' if submission of the revision is attempted 21 or more days after the deadline. If you do not think you will be able to meet this deadline please contact the editorial office immediately.

on behalf of Prof Mark Chaplain (Subject Editor)
openscience@royalsociety.org

Reviewer comments to Author:

Reviewer: 1

Comments to the Author(s)

The authors assess the impact of face masks on the control and mitigation of the COVID-19 pandemic in the US through dynamic models. The methodology is sound. The main concern is about two related model parameters c_m (the proportion of individuals who wear faces masks in public) and ϵ_m (the protective efficacy of face masks to prevent the transmission of infection to a susceptible individual). Although the values of these parameters are carefully chosen, these values were taken with no solid evidence. Considering the impact of these values on the quantitative results, it is important to provide more evidences/survey data on these values. Otherwise, the results would not be statistically sound.

Reviewer: 2

Comments to the Author(s)

See attached review report

===PREPARING YOUR MANUSCRIPT===

===PREPARING YOUR REVISION IN SCHOLARONE===

<https://royalsociety.org/journals/authors/author-guidelines/#supplementary-material> to include a suitable title and informative caption. An example of appropriate titling and captioning may be found at https://figshare.com/articles/Table_S2_from_Is_there_a_trade-off_between_peak_performance_and_performance_breadth_across_temperatures_for_aerobic_sc_ope_in_teleost_fishes_/3843624.

Author's Response to Decision Letter for (RSOS-210699.R0)

See Appendix B.

RSOS-210699.R1 (Revision)

Review form: Reviewer 1

Is the manuscript scientifically sound in its present form?

Yes

Are the interpretations and conclusions justified by the results?

Yes

Is the language acceptable?

Yes

Do you have any ethical concerns with this paper?

No

Have you any concerns about statistical analyses in this paper?

No

Recommendation?

Accept as is

Comments to the Author(s)

The authors have soundly addressed all concerns raised and the submission is acceptable in its current form.

Review form: Reviewer 2 (Salisu Garba)

Is the manuscript scientifically sound in its present form?

Yes

Are the interpretations and conclusions justified by the results?

Yes

Is the language acceptable?

Yes

Do you have any ethical concerns with this paper?

No

Have you any concerns about statistical analyses in this paper?

No

Recommendation?

Accept as is

Comments to the Author(s)

The manuscript has greatly improved and all suggested correction has been made.

Decision letter (RSOS-210699.R1)

Dear Dr Ngonghala,

It is a pleasure to accept your manuscript entitled "Assessing the impact of widespread respirator use in curtailing COVID-19 transmission in the United States" in its current form for publication in Royal Society Open Science. The comments of the reviewer(s) who reviewed your manuscript are included at the foot of this letter.

COVID-19 rapid publication process:

We are taking steps to expedite the publication of research relevant to the pandemic. If you wish, you can opt to have your paper published as soon as it is ready, rather than waiting for it to be published the scheduled Wednesday.

This means your paper will not be included in the weekly media round-up which the Society sends to journalists ahead of publication. However, it will still appear in the COVID-19 Publishing Collection which journalists will be directed to each week (<https://royalsocietypublishing.org/topic/special-collections/novel-coronavirus-outbreak>).

If you wish to have your paper considered for immediate publication, or to discuss further, please notify openscience_proofs@royalsociety.org and press@royalsociety.org when you respond to this email.

Due to rapid publication and an extremely tight schedule, if comments are not received, your paper may experience a delay in publication. Please note that your colleague's email address lmarinacci@partners.org is not currently receiving messages from the journal - please can you either confirm a new email address that is active or check with Dr Marinacci that the journal's email address is included on their 'white list' of emails.

on behalf of Prof Mark Chaplain (Subject Editor)
openscience@royalsociety.org

Associate Editor Comments to Author:

The reviewers are now satisfied your paper is ready for acceptance - congratulations!

Reviewer comments to Author:

Reviewer: 2

Comments to the Author(s)

The manuscript has greatly improved and all suggested correction has been made.

Reviewer: 1

Comments to the Author(s)

The authors have soundly addressed all concerns raised and the submission is acceptable in its current form.

Appendix A

Review of "Assessing the impact of widespread respirator use in curtailing COVID-19 transmission in the United States"

In this study, the authors designed a deterministic model to assess the impact of three types of face masks in controlling the COVID-19 pandemic in the United States. The authors used the cumulative mortality for the three waves of the COVID-19 pandemic in the US to estimate the unknown parameters of the model. Their simulations showed that respirators are far more effective in reducing COVID-19 burden when compared with the cloth masks or/and surgical/procedure masks. The authors also showed that the pandemic would have been prevented from being established in the US if 80% of the population in the US started wearing respirators during the first two months of the pandemic. The implication of this interesting result is that future pandemics of Coronaviruses or other influenza-like illnesses can be effectively halted from significantly taking off if respirators are stockpiled and made available to majority of the population. Hence, respirators can enable effective control of the pandemic before vaccines become available, which, collectively, mean we can potentially control such pandemics without having to shut down the economy. The manuscript is timely and well presented. Below are some of my specific comments or observations:

- (i) Adding the impact of human behaviour with regards to the use of face masks in the two-group model is interesting. Is there data to realistically estimate the values of the related parameters (α_{12} and α_{21})?
- (ii) One thing that may be relevant to the effectiveness of respirators or masks in general in reducing the burden of a pandemic of respiratory pathogen is the level of immunity in the community. What will be the minimum coverage needed, for the respiratory and surgical mask, for example, if half of the US population is immune due to immunization, which is the case at the present time, or natural recovery from previous infection with the SARS-CoV-2 virus? Will the coverage be reduced if a certain proportion of the population is already immunized? I understand the study is focussed on the early stages of a pandemic, and demonstrated that if we started using high quality masks at the early stages, the pandemic may not have taken off. Nonetheless, it is probably good to check whether the mask coverage needed for suppressing the pandemic may change as more people become immune to the virus.
- (iii) There are several respirators with different efficacies and uses. For clarity to the reader, I suggest the authors state the type(s) of respirators they are considering and provide the inward and outward efficacies of the mask considered in the study.

- (iv) In the formulation of the model, the difference between pre-symptomatic and asymptomatic infectious individuals is not very clear. I suggest the authors provide precise definition of the two classes, and also particularly indicate where individuals with mild symptoms of the pandemic belong to.
- (v) Since the pandemic trend (and data) suggests that the control reproduction number of the disease changes with time, the authors should provide the numerical value of the control reproduction number corresponding to each of the three pandemic waves in the US.
- (vi) In Section 2.2, the parameter c_r for the increase in the baseline values of the other NPIs needs to be further clarified. For instance, does $c_r = 0.05$ mean only 5% of US population improved their behavior to comply with effective face mask use?
- (vii) In the reference section: ref[36] is a repetition of ref[21].
- (viii) There are some typographical errors in the manuscript. The authors may wish to edit the manuscript more thoroughly to fix them. Below are a few suggested edits:
 - Line 195 Replace "in" with "as" in the phrase "...estimated baseline parameter values **in** presented in Table."
 - Line 544: Remove "of" in the subtitle "Supplementary Material: Formulation, Data Fitting and Parameter Estimation of **of** Extended Model"

In summary, the manuscript is interesting and addresses a major public health problem. A revised version of the manuscript that adequately addresses the above points is worthy of publication in the Royal Society Open Science.

Appendix B

Responses to Comments of Reviewers

We are very grateful to the reviewers for their very constructive comments. We have revised the manuscript accordingly. Below are our specific responses to the comments of the reviewers.

Responses to Comments of Reviewer #1

Comment: The authors assess the impact of face masks on the control and mitigation of the COVID-19 pandemic in the US through dynamic models. The methodology is sound.

Our response: We thank the reviewer for the summary and the comment on our methodology.

Comment: The main concern is about two related model parameters c_m (the proportion of individuals who wear faces masks in public) and \epsilonpsilon_m (the protective efficacy of face masks to prevent the transmission of infection to a susceptible individual). Although the values of these parameters are carefully chosen, these values were taken with no solid evidence. Considering the impact of these values on the quantitative results, it is important to provide more evidences/survey data on these values. Otherwise, the results would not be statistically sound

Our response: In line with the reviewer's comment, we have now provided additional evidence (including results from clinical trials) to support our choices for the values of the efficacy and face mask compliance parameters used in our simulations (see Table 1 of the revised manuscript). In particular, the mask efficacy parameter (\epsilonpsilon_m) is estimated from data provided in a laboratory evaluation of non-standard and N95 masks [37] and empirical studies in [38]. Further, the contour plot depicted in Figure 3 captures the robust and full-scale impact (and sensitivity) of these parameters on the disease dynamics (as measured by the reproduction number of the model). In other words, the contour plot allows us to evaluate the sensitivity of these parameters (for all possible values in their respective parameter space) on the reproduction number of the model. We have provided additional clarification about these in the revised manuscript.

Response to Comments of Reviewer #2

Comment: In this study, the authors designed a deterministic model to assess the impact of three types of face masks in controlling the COVID-19 pandemic in the United States. The authors used the cumulative mortality for the three waves of the COVID-19 pandemic in the US to estimate the unknown parameters of the model. Their simulations showed that respirators are far more effective in reducing COVID-19 burden when compared with the cloth masks or/and surgical/procedure masks. The authors also showed that the pandemic would have been prevented from being established in the US if 80% of the population in the US started wearing respirators during the first two months of the pandemic. The implication of this interesting result is that future pandemics of Coronaviruses or other influenza-like illnesses can be effectively halted from significantly taking off if respirators are stockpiled and made available to majority of the population. Hence, respirators can enable effective control of the pandemic before vaccines become available, which, collectively, mean we can potentially control such pandemics without having to shut down the

economy. The manuscript is timely and well presented. Below are some of my specific comments or observations:

Our response: We are very thankful to the reviewer for the nice summary of the manuscript.

Comment: (i) Adding the impact of human behaviour with regards to the use of face masks in the two-group model is interesting. Is there data to realistically estimate the values of the related parameters (α_{12} and α_{21})?

Our response: We thank the reviewer for this comment. Unfortunately, our extensive search reveal that no such data is currently available in the literature (this data can be obtained by following a cohort of individuals, some given masks and some not, over a long period of time). That's why we resorted to obtaining estimates of these parameters via fitting the model with observed epidemiological data.

Comment: (ii) One thing that may be relevant to the effectiveness of respirators or masks in general in reducing the burden of a pandemic of respiratory pathogen is the level of immunity in the community. What will be the minimum coverage needed, for the respiratory and surgical mask, for example, if half of the US population is immune due to immunization, which is the case at the present time, or natural recovery from previous infection with the SARS-CoV-2 virus? Will the coverage be reduced if a certain proportion of the population is already immunized? I understand the study is focused on the early stages of a pandemic, and demonstrated that if we started using high quality masks at the early stages, the pandemic may not have taken off. Nonetheless, it is probably good to check whether the mask coverage needed for suppressing the pandemic may change as more people become immune to the virus.

Our response: We thank the reviewer for this comment. Although, as noted by the reviewer, the focus of our study is to show the utility of respirators in effectively containing pandemics of respiratory infections if deployed early in the pandemics (before vaccines and other pharmaceutical interventions become available), we agree that the level of immunity later in the course of the pandemic may affect the coverage of the respirator (or mask in general) needed to effectively contain the epidemic. In line with the reviewer's comment, we checked the scenario where half the US population is immune (due to vaccination and/or natural recovery) and, as expected, the coverage level required for containing the pandemic in the US is significantly reduced. The utility and/or effectiveness of high quality masks (such as respirators) is far more effective when community transmissions are high (e.g., during each of the three pandemic waves in the US), and mask mandates are often relaxed when community transmission is low (which is the case now in the US, where, as noted by the review, at least half the population are immune due to vaccination.... and community transmission is a lot lower than during any of the three waves). In line with the reviewer's comment, we have provided further clarification on the fact that the mask coverage needed to effectively contain or suppress the COVID-19 pandemic is significantly reduced as more members of the community become immune (due to vaccination and/or natural recovery). Specifically, we have added a new section (Section 3.2.4 in the revised manuscript) that assesses the impact of vaccination and waning natural and vaccine-derived immunity on the burden of the pandemic. This entailed slightly modifying the basic model to allow for the implementation of a vaccination program and waning immunity, and carrying out additional numerical simulations.

We have added two new figures (Figures 6 and 7) in the main text and three new figures (Figures S3, S6, and S7) in the Supplementary Material to illustrate the results we obtained. Our simulation results show that waning of natural and vaccine-derived immunity could trigger multiple new waves of the pandemic in the US. The number, severity, timing of the peaks and duration of the projected waves depend on the quality of mask type used and the level of increase in the baseline levels of other NPIs used in the community during the onset of the third wave of the pandemic in the US. Specifically, no severe wave of the pandemic will be recorded in the US if immunity does not wane, or if surgical masks or respirators are used, instead of cloth masks (and particularly if the mask-use strategy is combined with an increase in the baseline levels of other NPIs). These results have enhanced the manuscript, and we are thankful to the reviewer for suggesting these.

Comment: (iii) There are several respirators with different efficacies and uses. For clarity to the reader, I suggest the authors state the type(s) of respirators they are considering and provide the inward and outward efficacies of the mask considered in the study.

Our response: We agree with the reviewer and we have clarified this fact in the revised manuscript, as suggested. In particular, we have added the following in the introduction: “Examples of NIOSH-certified respirators include N95, N99, N100, R95, P95, P99, and P100 particulate filtering face-piece respirators [14, 15]. Estimates for the efficiency of filtering face-piece respirators are nearly 100% for charged biological particles such as respiratory aerosols [15, 16,18]”

Comment: (iv) In the formulation of the model, the difference between pre-symptomatic and asymptomatic infectious individuals is not very clear. I suggest the authors provide precise definition of the two classes, and also particularly indicate where individuals with mild symptoms of the pandemic belong to.

Our response: We have provided additional clarification on the definition of the two classes in the revised manuscript.

Comment: (v) Since the pandemic trend (and data) suggests that the control reproduction number of the disease changes with time, the authors should provide the numerical value of the control reproduction number corresponding to each of the three pandemic waves in the US.

Our response: We agree with the reviewer, and we have provided the numerical values of the control reproduction number of the model, for each of the three pandemic waves in the US, in the revised manuscript (in particular, the values of the control reproduction numbers for Waves 1, 2, and 3 in the US are computed to be 3.13, 1.08, and 1.2, respectively).

Comment: (vi) In Section 2.2, the parameter c_r for the increase in the baseline values of the other NPIs needs to be further clarified. For instance, does $c_r = 0.05$ mean only 5% of US population improved their behavior to comply with effective face mask use?

Our response: The parameter c_r represents the additional reduction in the values of the community contact rate parameters from their baseline values due to further increases in the coverage and effectiveness of other nonpharmaceutical interventions. Thus, $c_r=0.05$ means the

further implementation of NPIs have succeeded in decreasing the baseline values of the contact rate parameters by 5%.

Comment: (vii) In the reference section: ref[36] is a repetition of ref[21].

Our response: We thank the reviewer for spotting this repetition, which we have now corrected.

Comment: (viii) There are some typographical errors in the manuscript. The authors may wish to edit the manuscript more thoroughly to fix them. Below are a few suggested edits: – Line 195 Replace "in" with "as" in the phrase "...estimated baseline parameter values in presented in Table." – Line 544: Remove "of" in the subtitle "Supplementary Material: Formulation, Data Fitting and Parameter Estimation of Extended Model".

Our response: We thank the reviewer for this comment and for the suggested edits.

Comment: In summary, the manuscript is interesting and addresses a major public health problem. A revised version of the manuscript that adequately addresses the above points is worthy of publication in the Royal Society Open Science.

Our response: We thank the reviewer for the comment.

Miscellaneous

In addition to addressing the comments of the two reviewers, we have carried out additional edits to further enhance the clarity of the manuscript. All major edits made (addressing reviewers' comments or the new results we added) are highlighted in blue in the revised manuscript.